# An in-situ polymerization strategy for gel polymer electrolyte Si‖Ni-rich lithium-ion batteries

Miao Bai[1], Xiaoyu Tang[1], Min Zhang[1], Helin Wang[1], Zhiqiao Wang[1], Ahu Shao[1] & Yue Ma [1] ✉

Coupling the Si-based anodes with nickel-rich $LiNi_xMn_yCo_{1-x-y}O_2$ cathodes (x ≥ 0.8) in the energy-dense cell prototype suffers from the mechanical instability of the Li-Si alloys, cathode collapse upon the high-voltage cycling, as well as the severe leakage current at elevated temperatures. More seriously, the cathode-to-anode cross-talk effect of transitional metal aggravates the depletion of the active Li reservoir. To reconcile the cation utilization degree, stress dissipation, and extreme temperature tolerance of the Si-based anode‖ NMC prototype, we propose a gel polymer electrolyte to reinforce the mechanical integrity of Si anode and chelate with the transitional cations towards the stabilized interfacial property. As coupling the conformal gel polymer electrolyte encapsulation with the spatial arranged Si anode and NMC811 cathode, the 2.7 Ah pouch-format cell could achieve the high energy density of 325.9 Wh kg⁻¹ (based on the whole pouch cell), 88.7% capacity retention for 2000 cycles, self-extinguish property as well as a wide temperature tolerance. Therefore, this proposed polymerization strategy provides a leap toward the secured Li batteries.

The electrification tide boosts the soaring market of electric vehicles (EV), unmanned aerial vehicles (UAV), high-end electronics, as well as distributed power supply solutions. Therefore, there is an urgent need for electrochemical cells that surpass the capabilities of current lithium-ion batteries (LIBs), as the energy densities of conventional rocking-chair formats (with $LiCoO_2/LiFePO_4$ cathodes and graphite anodes) have reached their ceiling limits[1–3]. Alternatively, the cell prototyping of the high-capacity silicon (Si) anode with the nickel-rich ($LiNi_xMn_yCo_{1-x-y}O_2$, NMC, x ≥ 0.8) cathode can achieve the enhanced energy densities of above 300 Wh kg⁻¹ at the cell level in theory[4,5]. When pairing the Si-based anode‖NMC model with conventional carbonate electrolytes, the electrode structure collapses during high-voltage cycling, leading to irreversible Li⁺ depletion as the cycle continues[6,7]. Besides, the dosage of flammable organic solvents poses severe safety risks like liquid leakage, thermal runaway, and even fire risks upon thermal shock or mechanical/electrical abuse scenarios[8,9].

In this regard, the reliable operation of the energy-dense battery system necessitates both the multiscale interfacial stability and controlled electro-chemo-mechanics of the electrodes, especially in harsh operating conditions (fast charging, high/low temperatures, etc.).

Featured with the high-capacity lithiation capability at room temperature (3579 mA h g⁻¹), appropriate equilibrium voltage (0.2–0.4 V *vs.* Li/Li⁺) and low cost ($4.20 ~ $6.50 kg⁻¹), Si materials triggered widespread interests both in the academic and industrial communities as the viable anode choice[10,11]. Unfortunately, the huge volume expansion upon the deep lithiation (>300%, $Li_{15}Si_4$) would lead to electrical contact loss and the gradual deactivation of the alloy particles, as illustrated in Fig. 1 (part i)[12]. So far, dimensional engineering (0D nanoparticles or 1D nanowires) or architectural innovation of the 3D Si structures have achieved performance progress at the low mass loadings when evaluated in half cells[13]. With the enhanced Si loading (>15-20 wt%) in the composite, unfortunately, the as-formed

¹State Key Laboratory of Solidification Processing, Center for Nano Energy Materials, School of Materials Science and Engineering, Northwestern Polytechnical University, Xi'an 710072, China. ✉e-mail: mayue04@nwpu.edu.cn

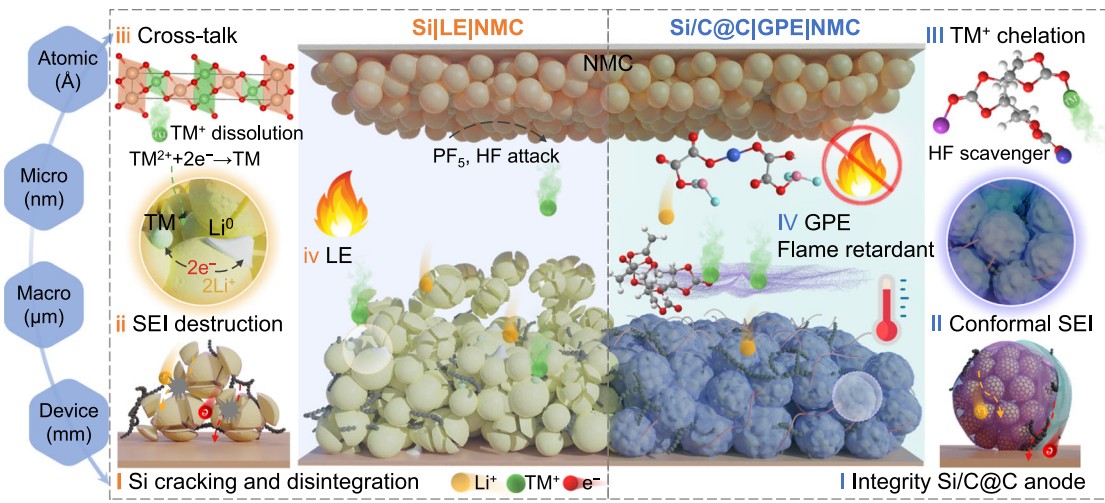

**Fig. 1 | Schematic illustration of Si-based anode‖NMC full cell model at multiple scales.** Left: the fading mechanisms of Si|LE|NMC prototype, which include: **i** the Si cracking and electrode pulverization; **ii** Li⁺ depletion caused by continuous SEI destruction and reformation; **iii** The SEI rupture and leakage current due to cathode-anode cross-talk effect; **iv** Safety hazards of liquid leakage and flammability. Right: The features of the as-proposed Si/C@C|GPE|NMC system, which include: **I** structure design of the Si/C@C anode; **II** the enhanced interfacial stability and mechanical robustness of the electrode upon the percolation with the nonflammable GPE; **III** the suppressed cross-talk effect; **IV** the flame retardancy of the GPE for the battery system.

solid electrolyte interphase (SEI) becomes mechanically fragile and insufficient to withstand the volume fluctuation during cycling (Fig. 1, part ii). Meanwhile, these interfacial reactions and the Li-Si intermediates irreversibly trap the active Li⁺, aggravating the cation depletion from the cathode at controlled areal capacity N/P ratio and lean electrolyte reservoir (~ 3 g Ah⁻¹ for the 18650 cells or even less for the pouch-format unit)[14]. Therefore, it is necessary to blend Si with graphite anodes while controlling the Si species content (<8–10 wt%), particularly at the areal capacity loadings > 3 mA h cm⁻² in Li-constrained cell systems[15]. Another noteworthy degradation mechanism of the Si-based‖NMC model lies in the cathode-to-anode cross-talk phenomenon[16]. Specifically, the acid species (HF) that derive from the Li salt would attack the layered oxide structure, meanwhile the transitional metal cations (e.g., $Co^{2+}$, $Ni^{3+}$, and $Mn^{3+}$) dissolve from the cathode lattice, then migrate and deposit onto the anode surface (Fig. 1, part iii), leading to the SEI fracture and accelerated self-discharge rates[17,18]. All these processes exacerbate the reaction irreversibility and result in rapid capacity fading, especially at elevated temperatures. Therefore, the harmony balance of the high-capacity merits and Li utilization degree is the prerequisite for the energy/power-dense Si-based anode‖NMC cell prototyping.

To stabilize the interfacial electrochemical process, fluoroethylene carbonate (FEC), Di-Fluoroethylene carbonate (DFEC) or methyl 2,2,2-trifluoroethylcarbonate (FEMC) additives have been introduced to derive the LiF species on the Si anode surface; while some Li-containing additives, such as lithium bis(oxalate) borate (LiBOB), lithium tetrafluoro(fluoromalonato) phosphate (LFMP) were explored as the sacrificial salts to construct the protective cathode electrolyte interface (CEI) on the Ni-rich oxide[19–23]. However, the continuous consumption of these additives upon cycling still prohibits the long-term viability of the electrodes, and the intrinsic safety concerns of organic electrolyte weeping and fire sensitivity remain unresolved (Fig. 1, part iv)[24]. In this regard, gel polymer electrolytes (GPEs) exploit the polymer matrix to immobilize solvent molecules for enhanced stability[25,26]. Various polymer matrices, including poly(methyl methacrylate) (PMMA) and poly(vinylidene fluoride-hexafluoropropylene) (PVDF-HFP), have been proposed as the cation carrier medium in the polymer lithium batteries[27–29]. For example, Song introduced cyanoethyl polyvinyl alcohol (PVA-CN) organogel electrolyte to stabilize

the interfacial electrochemical process of the Si anode (with loading mass of 1.3 mg cm⁻²)[30]. Huang et al. used the poly(poly(tetramethylene-ether) glycol-co-4,4'-methylene diphenyl diisocyanate)-ethylene diamine) as the GPE matrix, enabling the stable cycling of SiO| $LiNi_{0.5}Co_{0.2}Mn_{0.3}O_2$ cell (70.0% capacity retention (CR) after 350 cycles)[31]. Unfortunately, the prevailing discussion of GPE properties mainly focuses on the interfacial effects on the individual electrodes, while neglecting the synergistic action of multiple components or cross-over effect at the full-cell level. Furthermore, the enhanced mechanical strength of GPE networks always comes at the expense of the retard ionic diffusivity, especially in consideration of the insufficient ionic percolation among the particles[32]. To this end, the GPE design should simultaneously meet the following requirements: (1) adhesion strength and fracture toughness to tolerate the huge volume excursion; (2) the ionic diffusivity across the high-areal-capacity Si anode comparable to the liquid electrolyte (LE) system; (3) simultaneous construction of the stable SEI on the Si anode and CEI on the cathode surface; (4) suppression of cross-talk effect and leakage current; (5) flame retardant property. Thus far, the GPE design of such type has rarely been reported, let alone the mechanism elucidation on the system level.

Based on the aforementioned considerations, we combined the spatially arranged Si species within the carbon scaffold as well as the in-situ GPE polymerization strategy to dissipate the mechanical stress as illustrated in Fig. 1 (right part). The micron-sized Si and coal tar pitch (CTP) have been employed as low-cost precursors. In particular, the size range and dispersion of Si nanoparticles (Si NPs) in the composite (Si/C@C) were deliberately regulated (Fig. 1, part I). Meanwhile, a nonflammable, mechanical rigid GPE was formulated with lithium difluoro(oxalate) borate (LiDFOB) salt in cyclic poly(vinylene carbonate) matrix (PVCM) through an in-situ polymerization. Both finite element simulation and electrochemical evaluations suggest that the Si/C@C structure in conjunction with the mechanically reinforced, yet percolating PVCM-GPE would contribute to the effective stress dissipation without sacrificing the ion diffusion capability (Fig. 1, part II). The PVCM and DFOB⁻ species could chelate with transition metal ions to ameliorate cross-talk effect, enabling the multiscale interfacial robustness and 24% leakage current mitigation for the Si/C@C‖NMC811 prototype (Fig. 1, part III). As pairing the densely packed Si-

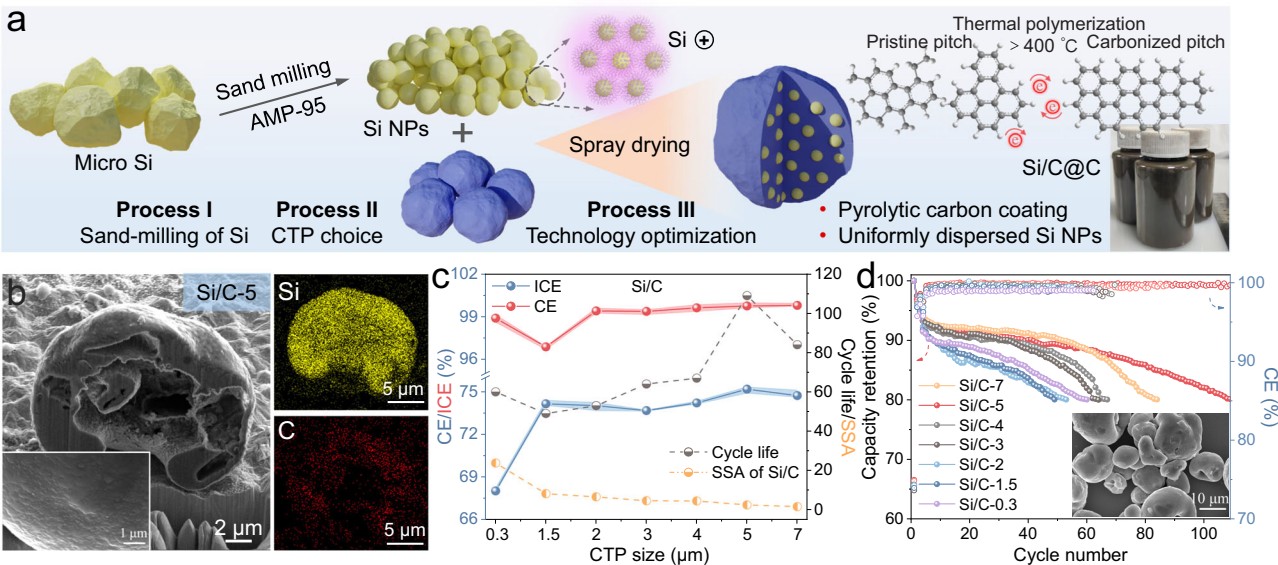

**Fig. 2 | Schematic illustration of the fabrication process of the Si/C@C composite. a** Process I: The particle size optimization of sand-milled Si; process II: the optimal CTP choice for the conformal Si encapsulation; process III: the spray drying and pyrolysis procedures. **b** The cross-sectional SEM image of the Si/C-5 composite processed by FIB and corresponding EDS elemental maps of C and Si. Inset is the enlarged image of the interfacial roughness. **c** SSA and cycle life (80% CR), average CE (average CE is the average of CE from the fourth cycle to a capacity retention rate of 80%) and initial CE (ICE) of Si/C with different CTP sizes (the shadows on the lines are pi error bars). **d** The CR of Si/C composites with different CTP sizes.

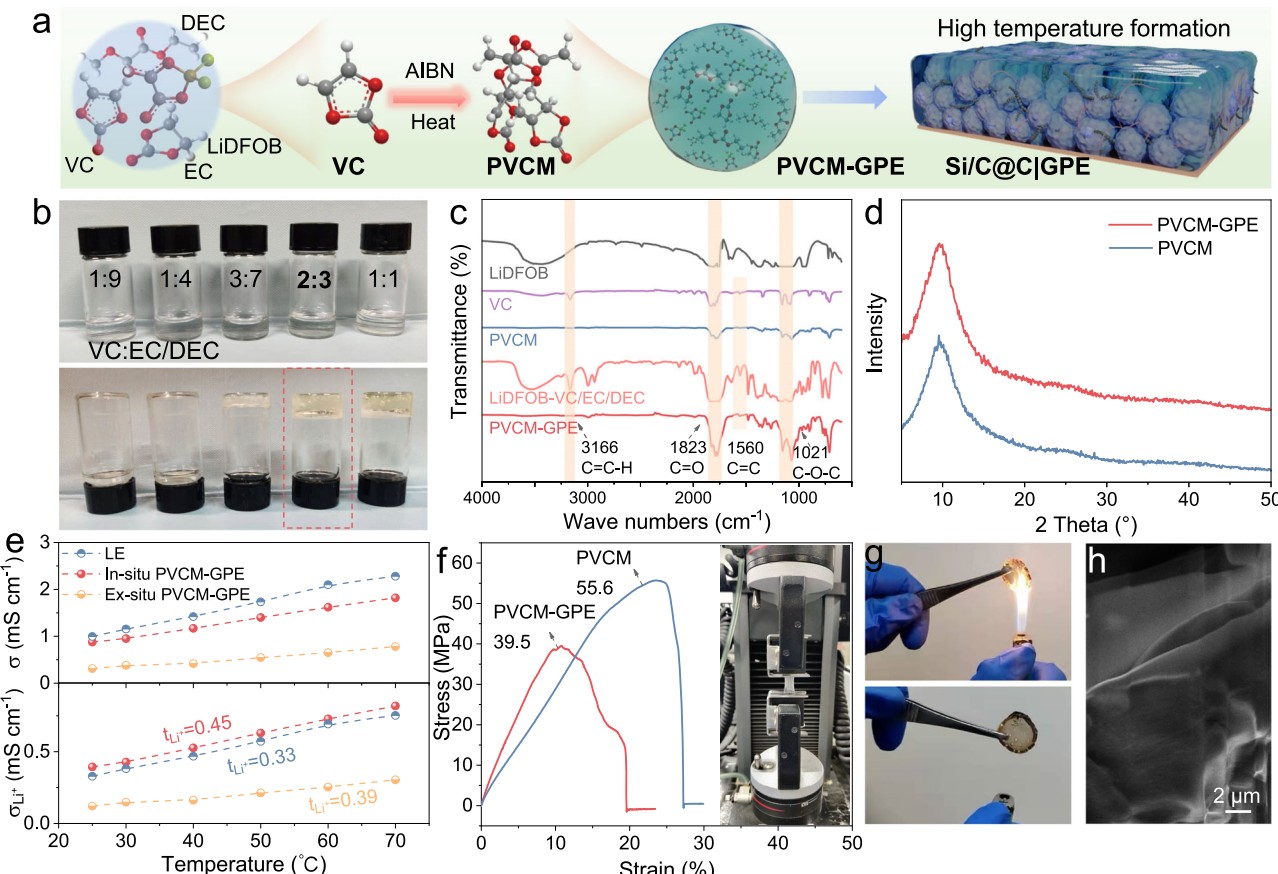

**Fig. 3 | Key characteristics of the PVCM-GPE. a** Schematic illustration of the preparation process of PVCM-GPE. **b** Optical photos of liquid electrolyte with varied amounts of VC before (top) and after (below) the polymerization process. **c** FT-IR adsorption spectra of LiDFOB, VC, PVCM, LiDFOB-VC/EC/DEC, and PVCM-GPE. **d** XRD patterns of PVCM and PVCM-GPE. **e** Temperature-dependent ionic conductivities and transference numbers of LE, in-situ PVCM-GPE, ex-situ PVCM-GPE. **f** Strain-stress curves of PVCM and PVCM-GPE. **g** The combustion experiment of PVCM-GPE. **h** SEM of PVCM-GPE after combustion.

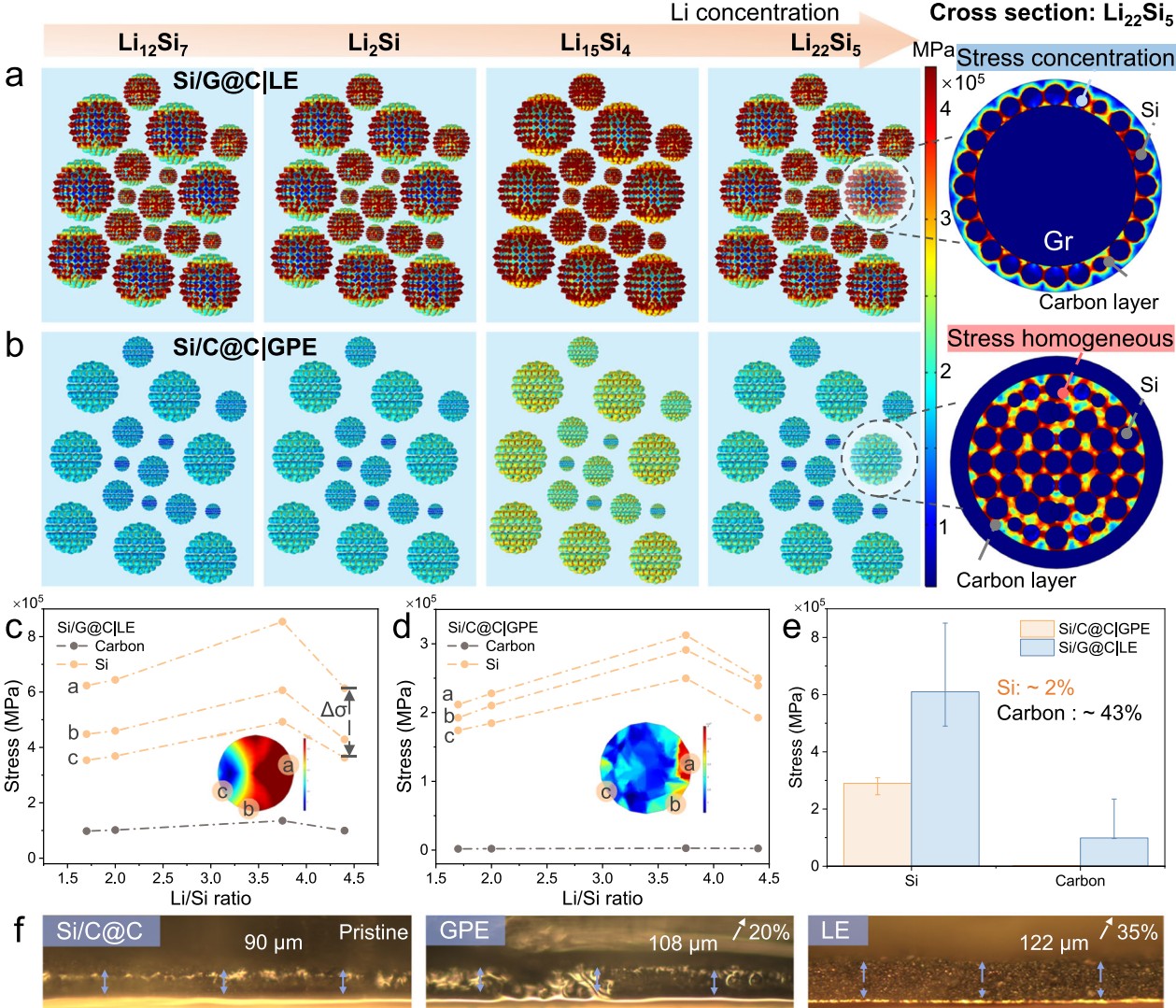

**Fig. 4 | Chemomechanical modeling of stress distribution during lithiation.** Stress distribution modeling across the anodes for **a** Si/G@C|LE and **b** Si/C@C|GPE at different lithiation states, and stress distribution of the single composite particle at the deep lithiation state. Stress evolution of Si NPs and carbon layer of **c** Si/G@C|LE and **d** Si/C@C|GPE. **e** Stress comparison of the Si and carbon layer species from two models. **f** Cross-sectional optical images of the Si/C@C electrode with GPE or LE before and after lithiation.

based anode and the NMC811 cathode in the GPE electrolyte model, the 2.7 Ah pouch-format cell rendered remarkable CR of 88.7% over 2000 cycles at 0.5 C, gravimetric energy density of 325.9 W h kg$^{-1}$ with the power density up to 1463.5 W kg$^{-1}$. In addition, this excellent temperature adaptability in the −20 to 60 °C range enables it to be deployed as a specialty battery.

## Results
The synthetic procedures of the hierarchical Si/C@C composite are schematically illustrated in Fig. 2a[33]. The preparation was initiated with a sand-milling process of Si microparticles (4 - 8 μm) into the nanoscale, where the time-dependent particle size distribution (PSD) was recorded in Fig. S1a (process I). After 2.5 h milling treatment, the PSD curve displays a median size value (D50) value of ~120 nm for the Si NPs, as confirmed by the field emission scanning electron microscope (FESEM) image shown in Fig. S1b. To avoid the spontaneous nanoparticle agglomeration, 2-amino-2-methyl-1-propanol (AMP) was deliberately added to modify the interfacial silanol groups[34]. Featured with the positively charged ammonium cation, as a result, the zeta potential of the Si NPs (1 M was evaluated as −5.7 mV) was altered to

+18.6 mV (1 M), enabling the monodispersity of Si NPs in the solution due to electrostatic repulsion. In the subsequent spray drying process (process III), the Si NPs were homogeneously dispersed within the pyrolytic CTP at the completion (Fig. S2).

To probe the spatial arrangement of the Si NPs in the composite, the focused ion beam (FIB) technique was employed to reveal the cross-sectional microstructural details. As the size of CTP precursor varies from 0.3 μm to 7 μm (Figs. 2b and S3), the as-obtained Si/C composites exhibit the rather loose (Si/C-0.3, 2.8 m$^2$ g$^{-1}$) or compact (Si/C-7, 1.6 m$^2$ g$^{-1}$) packing. Energy dispersive spectroscopy (EDS) elemental maps corroborate the evenly embedded Si NPs across the composite. A scrutiny of the high magnification SEM (inset in Fig. 2b) reveals the presence of the Si NPs within the pyrolyzed carbon layer. Transmission electron microscopy (TEM) image (Fig. S1d) and corresponding scanning TEM (STEM) coupled with energy dispersive X-ray spectroscopy (STEM-EDS) elemental maps (Fig. S3d) of the Si/C-5 composite suggest the conformal encapsulation of the Si NPs in the carbon coating layer. The pyrolytic carbon layer with highly graphitic ordering acts as the mechanical sheath against the volume expansion, meanwhile protecting the Si species from direct contact with the electrolyte.

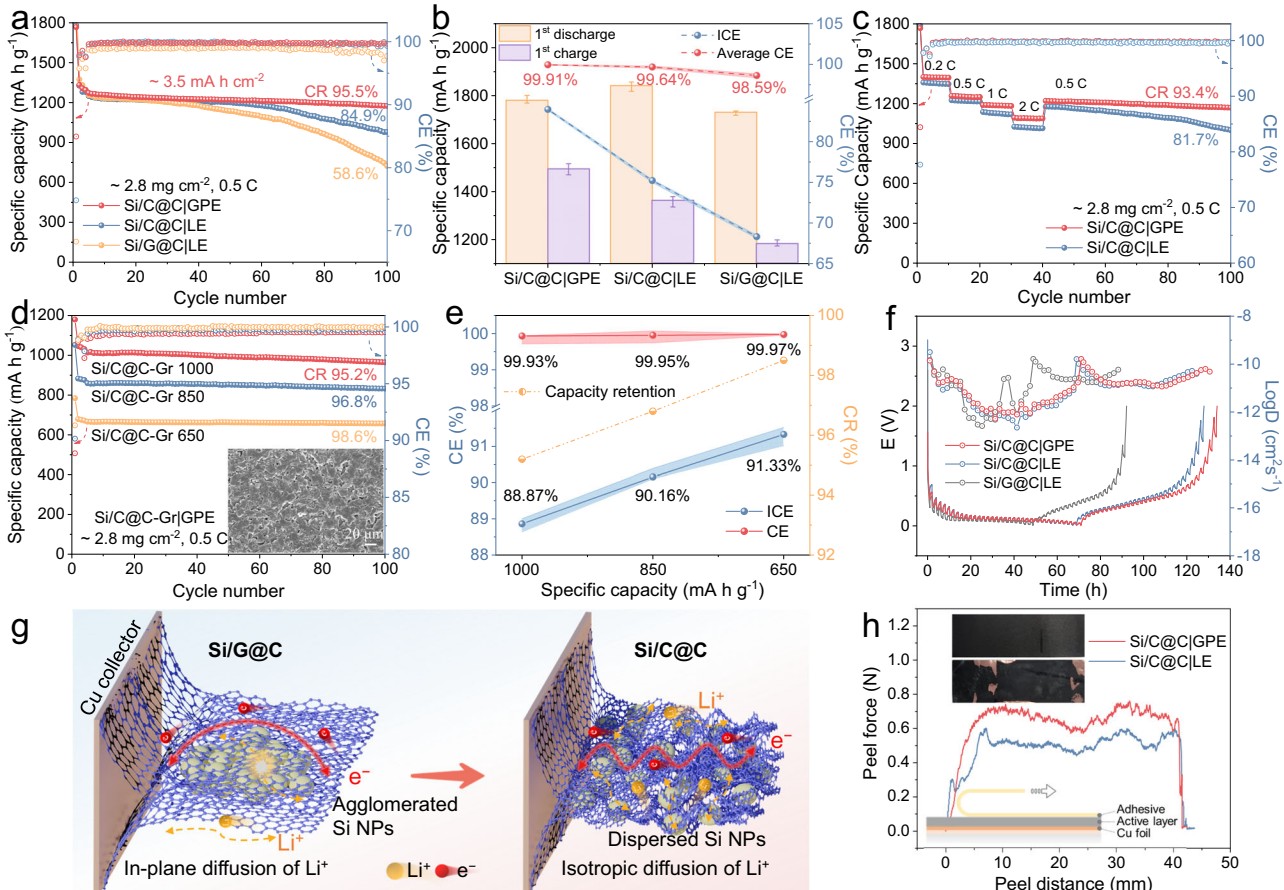

**Fig. 5 | Electrochemical performance of the Si-based composite anodes. a** The cyclability and CR values and **b** The first charge/discharge (pi error bar), ICE, and average CE of the Si/C@C|GPE, Si/C@C|LE, Si/G@C|LE anodes (-2.8 mg cm⁻²) at 0.5 C (1 C was defined as 1320 mA g⁻¹). **c** The rate behaviors of the Si/C@C|GPE and Si/C@C|LE anodes. **d** The cycle performance of the Si/C@C-Gr|GPE anode with pre-set nominal capacities at 0.5 C. **e** Summaries of CR, ICE, and average CE of Si/C@C-Gr|GPE. **f** The GITT curves of the Si/C@C|GPE, Si/C@C|LE, Si/G@C|LE anodes. **g** Schematic representations of Li⁺ migration pathways in Si/C@C and Si/G@C electrodes. **h** Peeling tests for the Si/C@C|GPE and Si/C@C|LE electrodes. The inset image in Fig. 5h: digital photographs of the electrodes after the peeling test.

Si/C composites with different CTP particle sizes were electrochemically compared using 1 M LiPF₆ in DEC:EC = 1:1 Vol% with 5% FEC as shown in Figs. 2c and S4. The charge/discharge process at 25 °C with the voltage range of 0.01 - 0.8 V at 0.5 C was conducted. Results show that the densely packed Si/C-5 and Si/C-7 composites with relatively smaller specific surface area (SSA) delivered enhanced Coulombic efficiency (CE) values. As shown in Fig. 2d, the Si/C-5 electrode exhibits the specific capacity of 1750 mA h g⁻¹ as well as a high average CE of 99.85%. The Si/C-5 was processed with the air jet milling and further coated with 5% CTP carbon. The as-obtained sample was designated as the Si/C@C. For comparison, Si NPs attached to the commercial graphite scaffold upon the spray drying procedure were investigated, the mass ratio of Si NPs to graphite in the precursor suspension was controlled as 1:1 (denoted as Si/G); in addition, Si/G@C product was also fabricated as the reprocessed product of the Si/G composite by CTP coating layer. The raw material optimization, compositional ratios and hierarchical microstructure features of these hybrid composites were elaborated in Fig. S5-8.

The in-situ polymerization of PVCM-GPE is schematically illustrated in Fig. 3a. In ethylene carbonate (EC)/diethyl carbonate (DEC) (1: 1, v/v) carbonate electrolytes, azobisisobutyronitrile (AIBN) additive could generate the free radicals upon heating to induce polymerization of vinylene carbonate (VC). Photographs of liquid electrolytes with the varied VC volume ratios before (top) and after polymerization (below) are shown in Fig. 3b. A condensed state of the GPE appeared as

the VC concentration exceeded 30%. To achieve the optimal balance between mechanical property and ionic conductivity, a 40% VC volume fraction was selected as the ideal formulation (Fig. S9a). Figure 3c compares the Fourier transform infrared spectroscopy (FT-IR) spectra of the LiDFOB salt in both the VC/EC/DEC mixed solvent and PVCM-GPE. The VC monomers exhibited the =C–H and C=C groups at ~ 3166 cm⁻¹ and 1560 cm⁻¹, which disappeared after the in-situ polymerization[35]. As compared with the monomer in the liquid electrolyte formulation (LiDFOB-VC/EC/DEC), PVCM-GPE displayed no obvious changes of the C=O vibration band at 1823 cm⁻¹ and the C–O –C vibration band at 1021 cm⁻¹, which implies that VC polymerization would not affect the chemical structure of O=C–O–C[36].

The X-ray diffraction (XRD) pattern of as-formed PVCM-GPE reveals a typical amorphous structure, suggesting the possible Li⁺ conduction along the polymer chains (Fig. 3d). The thermal behavior of the polymer electrolyte was investigated by thermogravimetry analysis (TG) and differential scanning calorimetry (DSC) (Fig. S9b). The DSC curves of the PVCM-GPE showed two small exothermic peaks centered at 120 °C and 148 °C, corresponding to the evaporation of residue DEC and EC, respectively. The rapid mass decay initiated at ~ 330 °C could be attributed to the thermal decomposition of the PVCM[37]. After storage at 80 °C after 24 h, furthermore, PVCM-GPE retained ~ 80% relative adsorption capacity of the electrolyte, demonstrating the excellent Li salt uptake capability of PVCM-GPE (Fig. S9c). Encouragingly, the PVCM-GPE exhibits ionic conductivity (σ)

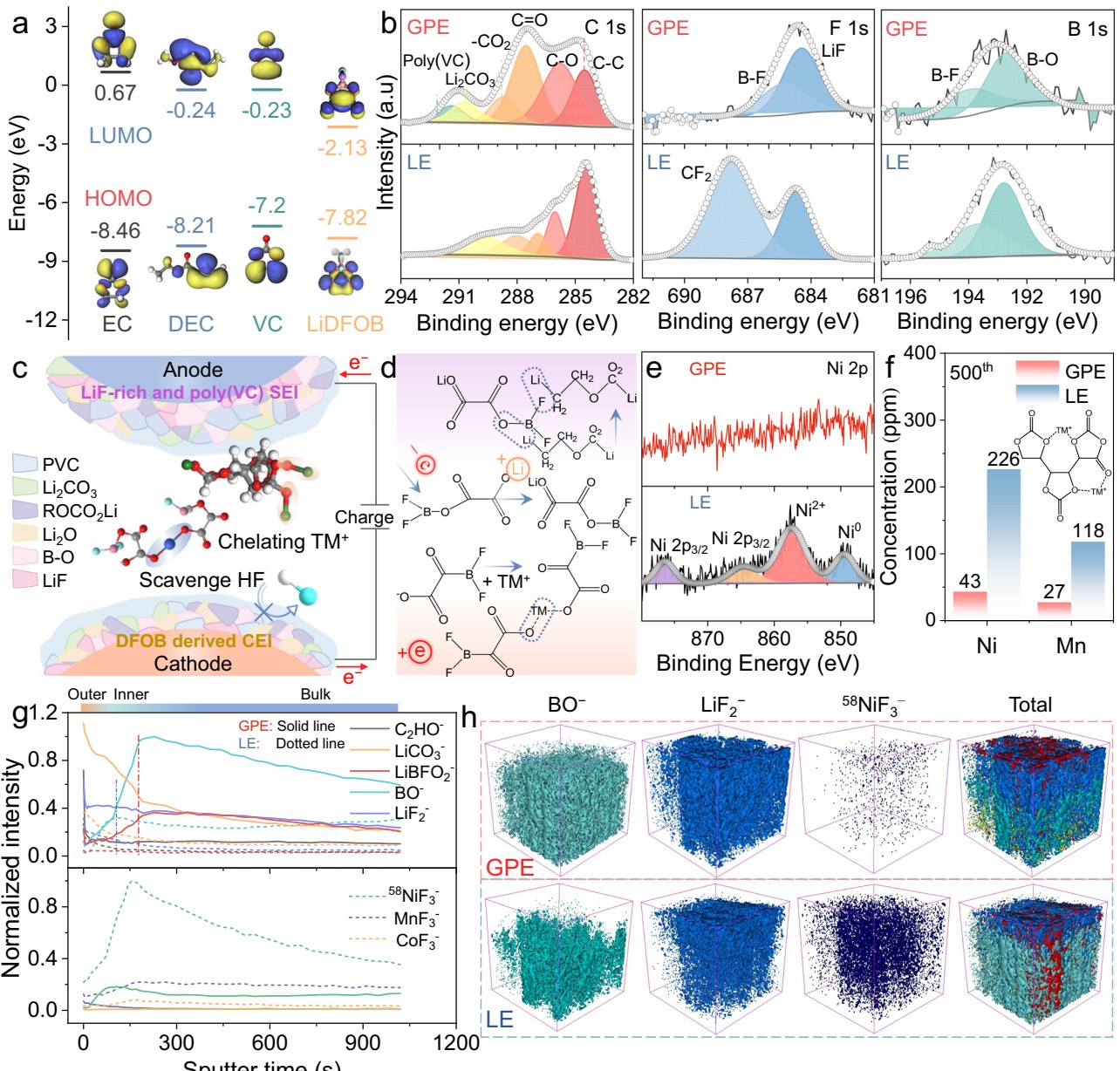

**Fig. 6 | Characterization of CEI and SEI in the Si/C@C|GPE|NMC811 full cell.**
**a** HOMO and LUMO energy levels of EC, DEC, VC, and LiDFOB interacting with Li⁺.
**b** The core-level C 1s, F 1s, and B 1s XPS spectra of NMC811 cathodes disassembled from the Si/C@C-Gr|GPE|NMC811 and Si/C@C-Gr|LE|NMC811 models after 100 cycles. **c** Proposed model of the SEI and CEI structure in the GPE on electrodes surfaces. **d** Chelation mechanism of TM cations with the LiDFOB salt involved in CEI layer. **e** Ni 2p XPS spectra, **f** ICP-MS analysis, and **g** TOF-SIMS depth profiles of selected fragments of the Si/C@C anode obtained from the Si/C@C-Gr|GPE|NMC811 and Si/C@C-Gr|LE|NMC811 models after 500 cycles. **h** TOF-SIMS 3D plots for $BO^-$, $LiF_2^-$, $^{58}NiF_3^-$ and total construction.

up to $8.61 \times 10^{-4}$ S cm⁻¹, which is in the same order of magnitude as the liquid electrolyte ($9.96 \times 10^{-4}$ S cm⁻¹) at 25 °C (Fig. 3e). According to the Bruce-Vincent-Evans equation, the Li⁺ transference number ($t_{Li+}$) of PVCM-GPE (0.45) is slightly higher than that of the liquid electrolyte (0.33) at 25 °C (Fig. S10a)[38]. The higher $t_{Li+}$ of PVCM-GPE should be mainly attributed to the strong electron withdrawing capability of C=O groups and the restrained anions movement[39]. Therefore, the sluggish anion mobility of PVCM-GPE was expected to enable less parasitic reactions at the electrode-electrolyte interfaces and the homogenized local electric field with reduced concentration polarization. PVCM-GPE exhibits a much higher onset potential of oxidative decomposition (4.8 V) as compared to that of the liquid electrolyte (4.5 V), suggesting

that the in-situ polymerization process has widened the electrochemical window (Fig. S10b).

Owning to the rigid cyclic carbonate backbone, pure PVCM exhibits a high tensile strength of 55.6 MPa; even with the addition of Li salt and plasticizer, the PVCM-GPE still maintained tensile strength value of 39.5 MPa[40]. In addition, the PVCM-GPE demonstrated immediate self-distinguishing property (1-2 s) when exposed to flame (Figs. 3g and S11)[41]. After the combustion process, the PVCM-GPE exhibits a dense carbon layer on the surface, which is expected to insulate the oxidizer and render the flame retardancy (Fig. 3h). Therefore, the in-situ polymerized PVCM-GPE balanced the properties of ionic conductivity, Li⁺ transference number, mechanical strength,

electrochemical stability with high voltage cathode and flame retardancy.

Based on material design strategies and the mechanical properties afforded by GPE, chemomechanical models were constructed via COMSOL software to estimate the mechanical stress distribution upon lithiation (Methods Section). The Solid Mechanics Physical Field Interface was used to simulate the mechanical response of the lithiated Si[42,43]. The stress tensor (σ) can be expressed as Eq. (1):

$$\sigma_{ij} = \frac{\upsilon E}{(1-2\upsilon)(1+2\upsilon)}\delta_{ij}\varepsilon_{kk} + \frac{E}{(1+\upsilon)}\varepsilon_{ij} \qquad (1)$$

where $\upsilon$ is Poisson's ratio; $E$ is Young's modulus; $\varepsilon$ is strain tensor[44]. The parameters of Li-Si alloys required in the simulation are given in Table S1. As shown in Fig. 4a, the Si/G@C electrode with LE (Si/G@C| LE) exhibited high stress concentrations in the Si particle region during the lithiation process. The Si NPs squeezed each other upon the deeper lithiation process, meanwhile the edges of Si particles were severely pulverized, eventually leading to the collapse of the electrode structure. The tensile stress and reverse compression at the point−point contact of the aggregated Si NPs suggest that the Young's modulus of the carbon layer is insufficient to stabilize the Si/G@C|LE model. In sharp contrast, as the Si particles were evenly distributed in the CTP derived pyrolytic carbon, the stress distribution became more homogeneous in the Si/C@C|GPE (Fig. 4b). The low stress concentration is attributed to the spatially confined individual Si NPs in the CTP-derived carbon. In addition, the elastic GPE exhibited a coupled effect in suppressing the mechanical stress.

By analyzing the stress change on the Si particle, a large stress gradient ($\Delta\sigma$) was observed near the point−point contact among the Si NPs, which would cause the particle crack (Fig. 4c). As compared in Fig. 4d, the carbon filler increased the interparticle contact area with the progressive stress dissipation, furthermore, the GPE encapsulation acts as a stress buffering layer to alleviate the volume expansion, which ensures the mechanical stability of the composite particles during the lithiation. As simulated with 3D modeling, there was little discrepancy in stress accumulation for the Si particle. The mechanical stress of Si in the Si/G@C|LE model is estimated ~2.1 times larger than in Si/C@C| GPE, while the compressive stress in the carbon layer is ~43 times as high (Fig. 4e). The dissipation of tensile stress ensures the mechanical stability of Si/C@C|GPE upon lithiation. Besides, the simulation models of Si/G@C|GPE and Si/C@C|LE have been demonstrated in Fig. S12.

Furthermore, the favorable effect of GPE on the structural robustness of electrode was assessed with in-situ optical microscopy. As shown in the cross-sectional views of the Si/C@C images before and after cycling (Fig. 4f), the Si/C@C|GPE electrode firmly anchored on the Cu foil and increased the electrode thickness from 90 μm to 108 μm at the deep lithiated state (100% SOC), corresponding to ~ 20% volume expansion. In stark contrast, the Si/G@C electrode in the LE exhibits a 35% increase. The above theoretical and experimental analysis thus suggests the mechanical flexibility and structural integrity of the electrode in contact with GPE.

Figures 5 and S13 summarized the electrochemical performance of the high-areal-capacity (~3.5 mA h cm⁻²) Si composite electrodes obtained in half-cells paired with GPE or LE (counter electrode in the half cell is Li foil). The first discharge and charge capacities of Si/C@C| GPE were documented as 1780 mA h g⁻¹ and 1590 mA h g⁻¹, rendering a satisfactory ICE of 89.35%. Furthermore, Si/C@C|GPE renders the best CR of 95.5% for 100 cycles and the average CE value (99.91%) from the 3ʳᵈ cycle onwards (Fig. 5b). The negligible capacity degradation demonstrates the structural robustness and the efficient Li⁺ utilization of the electrode. For comparison purposes, the Si/C@C|LE displayed ICE value of 75.19% and 84.9% CR value for 100 cycles. The inferior cyclability could be attributed to the possible detachment of Si NPs from the Gr flakes and the electrode collapse. The enlarged TEM image (Fig. S14a) confirmed that the Si NPs remained intimately encapsulated

in the pyrolytic carbon of Si/C@C composite without any fracture observed.

As compared in Fig. 5c, the Si/C@C|GPE maintained the specific capacity of ~ 1396 mA h g⁻¹ at 0.2 C and ~ 1091 mA h g⁻¹ at 2 C, indicating superior rate behavior as compared to the LE system. As shown in Fig. 5d - e, the CR values of Si/C@C-Gr|GPE with predetermined specific capacities of 1000 mA h g⁻¹ (denoted as Si/C@C-Gr 1000), 850 mA h g⁻¹ (denoted as Si/C@C-Gr 850) and 650 mA h g⁻¹ (denoted as Si/C@C-Gr 650) were evaluated as 95.2%, 96.8% and 98.6% for 100 cycles, respectively; the average CEs of all these hybrid anodes were maintained higher than 99.9% after 10 cycles. As shown in Fig. 5f, the Li⁺ diffusion coefficients ($D_{Li^+}$) of Si/C@C|GPE at lithiation state ranges from $1 \times 10^{-11}$ to $2 \times 10^{-9}$ cm² s⁻¹ and varies from $8 \times 10^{-10}$ to $3 \times 10^{-9}$ cm² s⁻¹ at delithiation state, which is approximately the same magnitude as the Si/C@C|LE electrode. The $D_{Li^+}$ of Si/C@C|LE ranges from $1 \times 10^{-12}$ to $4 \times 10^{-9}$ upon the full cycle. This difference is dictated by the different spatial configurations of the anode composites. As schematically illustrated in Fig. 5g, while the Li⁺ insert into graphite at different intercalation stages; the aggregated Si NPs posed an energy barrier for Li⁺ diffusion in the Si/G@C composite. In sharp contrast, the intrinsic microporous structure of the CTP derived from soft carbon would enable the facile Li⁺ migration in the Si/C@C model owning to the elimination of the intercalation stages[45]. Additionally, the increased number of prismatic surfaces enables a preferential de-solvation process of the Li⁺[46]. Since the tightly encapsulated pyrolytic carbon inhibits the Si NPs pulverization, the structural integrity also guarantees optimized CR even during the high-rate cycling[47].

In addition, peeling tests were performed to investigate the adhesion strength between the slurry-casting electroactive materials and the current collector. The peeling force of the Si/C@C|GPE electrode was evaluated to be 0.7 N, which is 28% stronger than that of Si/ C@C|LE electrode (0.5 N) (Fig. 5h). Apparently, the electroactive materials were firmly attached to the current collector without any cracks or exfoliation, indicating the robust structure of Si/C@C|GPE electrode.

Besides the intrinsic effects of the electrode design ingenuity and the GPE encapsulation, the performance deterioration on the cell level is also derived from the multiscale interfacial issues on the electrodes and the TM cation crossover from the cathode to anode. As shown in Fig. 6a, VC monomers exhibit higher highest occupied molecular orbital (HOMO) energy level than those of the EC and DEC, which implies the preferential reduction decomposition as compared to the solvent species; meanwhile the LiDFOB salt exhibits the lower lowest unoccupied molecular orbital (LUMO) and thus can be oxidized on the cathode electrode surface to form the protective CEI layer[48]. Figure S16 presents the surface topography of the post-cycled electrodes retrieved from the Si/C@C-Gr|GPE|NMC811 model (0.5 C between 4.2 and 3.0 V after 100 cycles), which shows the uniform SEI and CEI layers on the Si/C@C and NMC811 electrodes with the characteristic F and B elements distributed across the interfaces. More evidence of effective LiDFOB-GPE protection can be obtained from the high-resolution X-ray photoelectron spectroscopy (XPS) spectra. As expected, the CEI is mainly covered by the in-situ polymerization products of poly(VC) (C 1 s, 291.3 eV) as compared to its counterpart in LE (Si/C@C-Gr|LE| NMC811) (Fig. 6b).

In addition, the LiDFOB tends lose electrons on the oxide surface upon charging, resulting in the ring-opening reaction of the difluoroborane (HBF₂) radical. This radical further reacts with the EC solvent and forms a series of polycarbonate like oligomers on the cathode surface (−CO₂ at 288.5 eV) (Fig. 6c, d)[49]. The LiF (F 1 s, 684.8 eV), B−O (BₓOᵧ species, B 1 s 192.4 eV), B−F (B 1 s, 193.5 eV) species are derived from LiDFOB decomposition, which effectively suppresses further electrolyte corrosion and mitigates the parasitic reactions. For the O 1 s spectra, C=O bond situated at 532.4 eV and ROCO₂Li at 533.7 eV were also observed, suggesting EC/DEC molecules are probably involved in

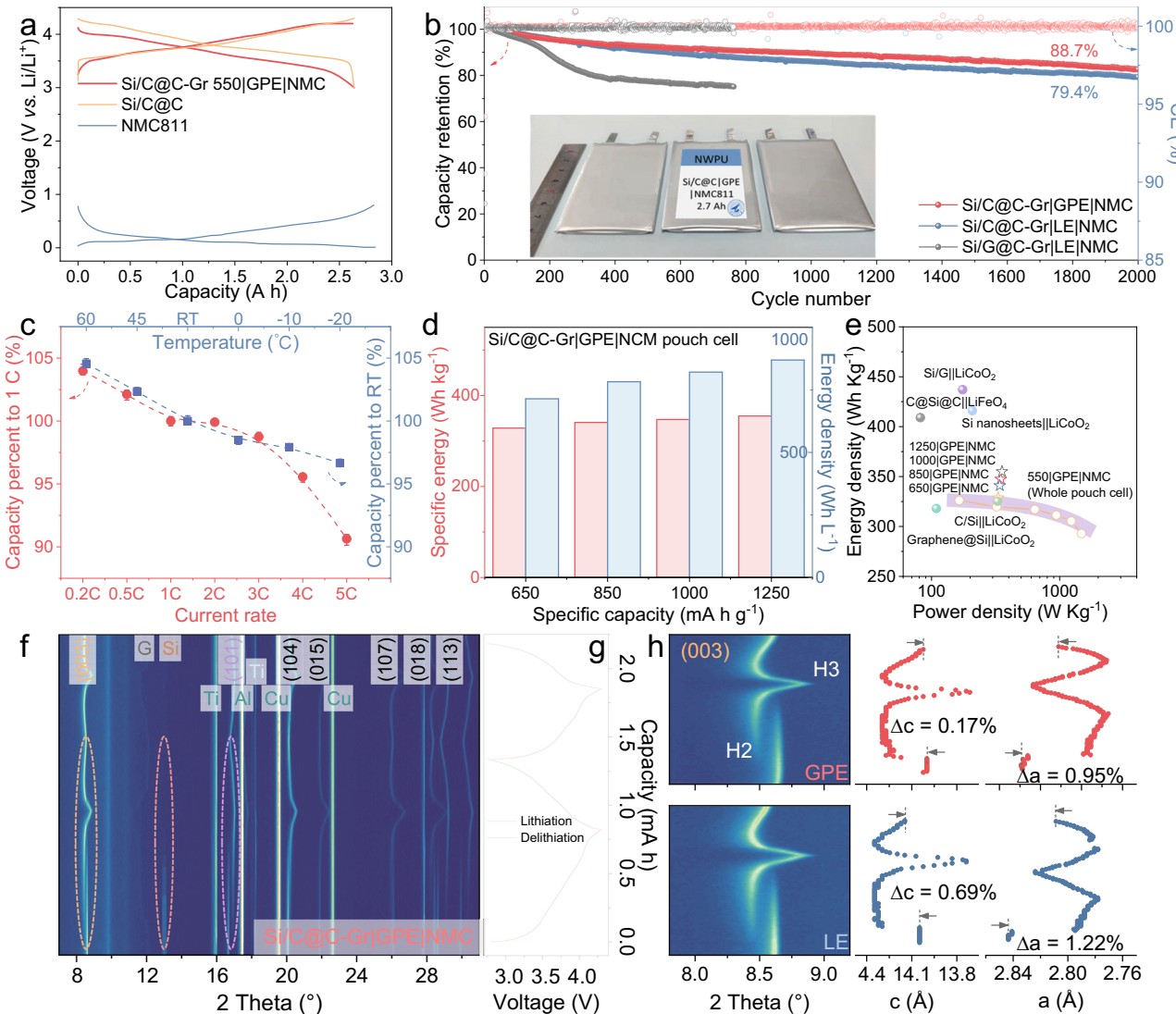

**Fig. 7 | The electrochemical performance of the Si/C@C-Gr|GPE|NMC full cell.** **a** Charge and discharge profiles of the Si/C@C-Gr 550 anode, NMC cathode, and Si/C@C-Gr 550|GPE|NMC pouch-format cell. **b** Cycling behavior of the full cells. **c** The rate behaviors and wide-temperature-range performance of the Si/C@C-Gr|GPE|NMC full cell. **d** Gravimetric/volumetric energy densities of Si/C@C-Gr 650, Si/C@C-Gr 850, Si/C@C-Gr 1000, and Si/C@C-Gr 1250 anodes coupled with NMC in GPE. **e** Ragone plot in comparison with energy densities at maximum power densities from previous literature. **f** Contour plots of operando XRD spectra and **g** the voltage profiles of Si/C@C-Gr 1000|GPE|NMC. **h** The corresponding shift of (003) reflections and calculated lattice parameters along c-axis and a-axis of NMC811 are retrieved from Si/C@C-Gr 1000|GPE|NMC and Si/G@C-Gr 1000|LE|NMC, respectively.

the CEI formation (Fig. S17)[50]. The difference in the O1s spectra lies in the pronounced sub-peak at 529.4 eV for the Si/C@C-Gr|LE|NMC811 model, which originated from the typical metal oxide (M–O) bond. This comparison also indicates the incomplete passivation of the NMC cathode in the LE system[51,52]. As compared to the cycled anode in the LE system, similarly, the SEI layer at the Si/C@C-Gr|GPE interface contains more poly(VC) (C 1 s, 291.3 eV), yet obviously mitigated Ni–O bonds (O 1 s 529.4 eV) (Fig. S18). Apparently, the presence of abundant charged carbonyl group (C=O) in PVCM effectively stabilized cathode surface by forming −O=CO...TM chemical bond with layered oxide particles. Simultaneously, $F_2BO^-$ in LiDFOB chelated with the TM cations and forms an insoluble protective film on surface[53–55]. In addition, LiDFOB salt supplies electron-deficient boron atoms that can scavenge HF and trace amounts of moisture in the electrolyte[56–58]. For the fully delithiated anode as paired with the GPE, the SEI of the anode exhibits much weaker Ni signals than the anode paired with pristine LE (Fig. 6e). Accordingly, the inductively coupled plasma mass spectrometry (ICP-

MS) analysis showed that 226 ppm of Ni ion and 118 ppm of Mn ion were observed at the Si/C@C anode interface in contact with the LE; while only 43 ppm of Ni ion and 27 ppm of Mn ion were observed at anode in GPE, which illuminated that the largely suppressed cross-over effect of TM dissolution from the NMC cathode (Fig. 6f).

To gain more insights into the spatial arrangement of the SEI layer, time of-flight secondary-ion mass spectrometry (TOF-SIMS) was employed to analyze the depth-resolved compositional features on the Si/C@C anode that both cycled in GPE and LE. At the surface, the fragment contents of $C_2HO^-$ (representing organic species) and $LiCO_3^-$ species became gradually pronounced, indicating that the organic layer composed of alkoxy carbonate formed at the interface (Fig. 6g)[59]. Noted the slightly higher intensity of $C_2HO^-$ in GPE was observed as compared to the SEI formed in LE model, which could be attributed to the reductive deposition of PVCM species. Along the sputtering depth, the content of $LiBFO_2^-$, $BO^-$, and $LiF_2^-$ fragments (representing inorganic constituents related to LiDFOB decomposition) continuously

increased, meanwhile the LiCO$_3^-$ content sharply dropped. Furthermore, LiF$_2^-$ was only dominantly present in the electrode outer layer in contact with the LE, whereas LiF$_2^-$ exists in both the outer and inner (together with the BO$^-$ species) layers of the SEI in Si/C@C-Gr|GPE system. This inorganic-rich composition would enhance the mechanical robustness and ionic conductivity of the SEI layer, stabilizing the interfacial electrochemistry of the active Si anode. Meanwhile, the cross-over of the dissolved TM$^+$ on the surface of the anode electrode is effectively reduced when pairing with the GPE, as evidenced by much weaker $^{58}$NiF$_3^-$/MnF$_3^-$/CoF$_3^-$ signals of the SEI layer. Consistent with the ICP-MS results, this phenomenon further validated the protective role of LiDFOB-PVCM-derived CEI for the cathode (Fig. S19). As compared with the Si/C@C-Gr|LE|NMC811 model, the much lower leakage current value of Si/C@C-Gr|GPE|NMC811, indicates the suppressed electron-transfer dynamics and the inhibited parasitic interfacial reactions (Fig. S20)[60]. Compared with the traditional SEI layer in LE system, 3D views of interfacial species for the Si/C@C-Gr|GPE directly visualize the homogeneous and mosaic stacked layers including the organic species, BO related species and inorganic LiF species conformally covered on the electrode, therefore the highly reversible alloy/dealloying reactions can be expected (Fig. 6h).

Through the in-situ GPE polymerization strategy during the electrolyte injecting and formation process, a 2.7 A h pouch-format full cell of Si/C@C-Gr|GPE|NMC was assembled. Figure 7a exhibits the discharge-charge curves of the Si/C@C-Gr 550 anode (blend with the Gr anode to achieve the nominal capacity of 550 mA h g$^{-1}$), the NMC cathode, and Si/C@C-Gr 550|GPE|NMC full cell. The Si/C@C-Gr 550|GPE|NMC exhibited an initial charge capacity of 2.69 A h and discharge capacity of 2.53 A h, respectively, corresponding to an ICE of 94.21% at 0.5 C. A stable CR of 88.7% was achieved for 2000 cycles, in contrast to merely 79.4% CR of the Si/C@C-Gr 550|LE|NMC and the rapid capacity decay of the Si/C@C-Gr 550|LE|NMC after 300 cycles (81.4%) (Figs. 7b, S21). As shown in Fig. S22, the thickness of the Si/C@C-Gr 550|GPE electrode increased from 110 μm (double-side coated, comprising 12 μm Cu foil thickness) to 126 μm at 100% state of charge (SOC) after 100 cycles, corresponding only ~ 17% volume expansion.

Benefitted from superior ionic and electronic conduction, as shown in Fig. 7c, the Si/C@C-Gr 550|GPE|NMC full cell retains discharge capacities of 103.9%, 102.1%, 99.2%, 97.4%, 95.5%, and 91.6% compare with theoretical capacity at 0.2 C, 0.5 C, 2 C, 3 C, 4 C and 5 C, respectively. What's more, the performance at wide-temperature-range was evaluated, Si/C@C-Gr 550|GPE|NMC possesses discharge capacities of 104.5%, 102.3%, 98.5%, 97.9%, and 96.7% of 25 °C performance at 60 °C, 45 °C, 60 °C, 0 °C, −10 °C and −20 °C, respectively. In an attempt to probe the energy density, we assembled full cells using different mass ratios of Gr blended anodes, as shown in Fig. S23. As the specific capacity increased from 550 to 650, 850, 1000, 1250 mA h g$^{-1}$, the Si/C@C-Gr|GPE|NMC pouch-format prototype is capable of delivering CR of 98.1%, 97.2%, 95.2%, 93.3%, 89.8%, meanwhile, gravimetric energy densities of 325.9, 328.5, 340.9, 347.4, and 355.0 W h kg$^{-1}$, respectively (based on the whole pouch cell) (Fig. 7d, Table S2). We calculated the volumetric energy density as 846.8 W h L$^{-1}$ based on the dimension of the pouch full battery. Figure 7e exhibits the Ragone plot of the Si/C@C-Gr|GPE|NMC prototype in comparison with the Si-based full cell models from the literature (based on the active material). The energy density of 292.7 W h kg$^{-1}$ can be achieved at the maximum power density of 1463.5 W kg$^{-1}$ for Si/C@C-Gr 550|GPE|NMC (Table S3).

Furthermore, transmission mode XRD was employed to document the real-time phase evolution of the Si/C@C-Gr 1000 anode and NMC811 cathode in a single-layer pouch cell with the Mo-Kα radiation source (Figs. S24, 7f). The diffraction pattern exhibited strong characteristic peaks that could be indexed to (003), (101), (104), (105), (107) and (113) reflections of layered NMC 811 structure, while the (002) Gr peak and (111) cubic Si crystalline of the Si/C@C-Gr anode are also demonstrated in the contour plot. For the anode electrode, the (002)

Gr peak at 12.1° shifted to 11.5° upon the Li$^+$ intercalation to form the LiC$_{12}$, subsequently towards a lower value (11.0°) at the deep charge state, corresponding to the LiC$_6$ (001) plane. With the progress of charging, the peak shifted to a higher angle and gradually returned to the original position. At the meantime, the peak intensity of the crystallized Si (13.1°) gradually weakened and disappeared until the lithiation process till 0.01 V, suggesting the amorphization process of Si. As for the NMC811 cathode, the (003) peak at 8.6° is attenuated and an alternative peak appeared at a lower 2θ angle during the charging process, indicating a discontinuous phase transition from the H1 phase to the H2 phase and expansion of the c-axis (Fig. 7h). At the end of the charging process, the (003) peak drastically returned to higher 2θ (8.8°) that indexed to H3 phase[61]. As for the cathode in the Si/C@C-Gr 1000|GPE|NMC811 pouch cell, the c value exhibits a pronounced decreased from initial 13.99 Å during the charge process and then almost recovered to the original 14.02 Å during the discharge process, corresponding to a negligible variation along (001) axis with Δc = 0.17%. In sharp contrast, the NMC811 cathode experienced an obvious lattice variation along the c axis (Δc = 0.69%) in Si/G@C-Gr 1000|LE|NMC811, demonstrating a more irreversible phasic variation with Li$^+$ consumption. Simultaneously, Δa of NMC811 in GPE (0.95%) is also lower than that of in LE (1.22%). As described in Figure. S25, the inductively coupled plasma (ICP) technique provides Li quantifications of both the anode and cathode. After 300 cycles, the Li content in cathode at 0% SOC drops to 5.16% in the Si/C@C-Gr 1000|GPE|NMC811 model, with 2.11% remaining in the anode electrode; the opposite trend is observed in the Si/G@C-Gr 1000|LE|NMC811 model as a control sample (4.63% of cathode and 2.75% in anode). This indicates that the Li$^+$ in the Si/C@C-Gr 1000|GPE|NMC811 prototype can be efficiently utilized under the synergistic effect of electrode design innovation as well as the interfacial stability, thus extending the cycle life of the cell model.

## Discussion

In summary, we systematically unraveled the degradation origins of the high-capacity Si anodes in the energy-dense cell prototype. Correspondingly, a robust Si/C@C composite design with the combined GPE percolation network was proposed, which demonstrated mechanical robustness of the electrode at the high-areal-capacity loading. Meanwhile, the highly elastic PVCM matrix incorporated the LiDFOB salt, which balanced the high-rate ionic conductivity as well as the stabilized solid electrolyte interface in contact with the Si-based anode and cathode electrolyte interface with the NMC cathode, respectively. This nonflammable GPE effectively suppressed continuous electrolyte decomposition and TM cation crossover effect, guaranteeing enhanced thermal stability and secured operation of the battery system. The 2.7 Ah pouch-format cell was assembled by integrating the Si/C@C-Gr anode and NMC cathode with the PVCM-GPE, the model of which achieved robust CR (88.7% over 2000 cycles), wide-temperature adaptability (−20-60 °C) as well as the gravimetric energy densities of 325.9 W h kg$^{-1}$ at the extreme power output of 1463.5 W kg$^{-1}$. The insight from this study not only elucidates the multiscale fading mechanisms of the energy-dense prototypes, but also motivate the in-situ polymerization approaches to overcome critical safety issue of the Si/C@C|NMC811 model, especially as application requirements moves toward energy/power-dense regimes.

## Methods
### Material synthesis
Preparation of the Si/C@C, Si/G@C, and Si/G composites. All materials were used as received, unless stated otherwise. In a typical synthesis, firstly, commercial micron-sized silicon powder (Si, 3-5 μm, Hefei Kell Nano Energy Technology Co., Ltd.) was mixed with alcohol (99.9%, Aladdin) (20% solid content). The wet sand milling was conducted for 1-2.5 h using a rotation mixer at 1100 rpm (WSP-10, Longxin Intelligent

Equipment Co., Ltd), while 0.5% AMP-95 (95% 2-amino-2-methyl-1-propanol in an aqueous solution, Sinopharm Chemical Reagent Co., Ltd.) was introduced into the mixture as dispersant. Then the coal tar pitch (CTP, Shanxi Coking Coal Group Co., Ltd) was added into the above dispersion, followed by mechanical stirring for 1 h (mass ratio of Si to CTP is 1:1). The Si/CTP microspheres were prepared by spray drying process. After that, Si/C composite was obtained by a sintering process at 920 °C for 3 h under nitrogen atmosphere, in which the heating rate was 7 °C min⁻¹. In further, to improve the compaction density of the electrode, crushing is performed by air jet milling and further by tar coating. The obtained product was denoted as Si/C@C.

Fabrication of polymer gel electrolyte. The flexible PVCM-GPE with tunable size was fabricated by in-situ polymerization. Firstly, the solutions of 1 M LiDFOB in VC and 1 M LiDFOB in ethylene carbonate/dimethyl carbonate (EC/DEC, 1/1, v/v) were prepared under argon atmosphere. Among them, the mixed solution of VC and EC/DEC was prepared in different volume ratios of 1: 9, 1: 4, 3: 7, 2: 3, and 1:1, respectively. Subsequently, the azobisisobutyronitrile (0.2 wt %) (AIBN, vs VC) as the catalyst was added into the mixed solution and stirred for 2 h. The electrolyte solutions were stored at 60 °C for 24 h and followed by the polymerization at 80 °C for another 2 h to obtain PVCM-based gel polymer electrolytes (named PVCM-GPE). The procedure was kept and handled in a glovebox circulated with high-purity argon gas (<1 ppm $O_2$ and <1 ppm $H_2O$).

For reference, the liquid electrolyte (LE) designed in this paper is 1 M LiDFOB in ethylene carbonate/dimethyl carbonate (EC/DEC, 1/1, v/v).

## Material characterization and instruments

The morphologies of all samples were observed by field-emission scanning electron microscopy (FESEM) (FEI, Nova Nano SEM 450) and a transmission electronic microscope (TEM) (FEI, Talos F200X TEM). Focus ion beam (FIB) profiling and observation of the samples were performed using a Helios G4 CX dual-beam instrument. Particle size distribution (PSD) was conducted on a Malvern Mastersizer 3000 laser particle size distribution analyzer. Brunauer-Emmett-Teller (BET) analysis was conducted via nitrogen adsorption/desorption isotherms at 77 K, the data was collected using a micromeritics ASAP 2460 system. XRD analysis was performed in a transmission mode X-ray diffractometer (STADIP STOE) with a position-sensitive detector and Mo Kα1 radiation with the wavelength (λ) of 0.7107 Å, operating at 50 kV and 40 mA. Thermogravimetry analysis (TG) measurements were carried out with a heating rate of 10 °C/min using Mettler Toledo TGA-DSC 3+ instrument. Raman spectrums were performed on a Raman spectrometer (HORIBA, France) with 532 nm line of a helium-neon as the excitation beam. The Kratos Axis Supra X-ray photoelectron spectroscopy (XPS) system offers surface analysis. The Fourier transform infrared (FT-IR) spectroscopy measurements of samples were measured on a Nicolet Nexus 5700 (Thermo Electron Corporation, USA). Zeta potentials were measured on a Zetasizer 2000 (Malvern) instrument. The amount of transition metal deposits on the cycled anode was measured by inductively coupled plasma mass spectrometry (ICP-MS) (Aglient 7800). Peeling tests and tensile experiments were carried out on CMT4204 electronic universal material testing machine. The anodes were cut into 1 × 3 cm² specimens and 3 M tape was attached to each specimen for peeling tests. The peeling speed was 100 μm s⁻¹, and the peeling strengths during the detachment were recorded. Elemental analysis and depth profiles of the SEI were determined by time-of-flight secondary ion mass spectroscopy (TOF-SIMS) (PHI nano TOF II). The size of the TOF-SIMS sputtered area is 100 μm × 100 μm. Ion etching was performed by Cs⁺ ion beam with a current ~ 2 nA within 30 s in each cycle. The depth profiles were processed and analyzed by TOF-DR software.

## Electrochemical measurements

PVCM based gel polymer electrolyte. For the PVCM-GPE, the electrochemical stability windows were determined from linear sweep voltammetry (LSV) measurement using stainless steel as the working electrode and Li as the counter and reference electrode. The test was performed at a scan rate of 1 mV s⁻¹.

The liquid electrolyte uptake of PVCM was obtained by measuring the mass of PVCM before and after immersing in electrolyte for 10 h, and calculated according to the following Eq. (2):

$$\Delta m(\%) = \frac{m_1 - m_0}{m_0} \times 100\% \tag{2}$$

where $\Delta m$ represents the electrolyte uptake (%), $m_O$ and $m_1$ are the mass of the PVCM before and after immersing in liquid electrolytes, respectively.

The ionic conductivity ($\sigma_i$) of GPE was calculated based on the bulk resistance measured by AC impedance test applied for SS|GPE|SS cells, using the following Eq. (3):

$$\sigma_i = \frac{L}{R \times S} \times 100\% \tag{3}$$

where $L$ is the thickness of GPE, $S$ represents the contact area between stainless steel (SS) and GPE, $R$ is the measured resistance.

The Li ion migration number ($t_{Li^+}$) of GPE was obtained by chronoamperometric method and EIS method. Before measurement, the in-situ GPE was assembled into Li symmetric cell and rested for 12 h. Then, the obtained Li|GPE |Li cells were used to perform potentiostatic polarization test at applied voltage of 10 mV. The EIS measurements were performed in the frequency range from 1 to 10⁶ Hz. The $t_{Li^+}$ was calculated from BruceVincent-Evans equation:

$$t_{Li^+} = \frac{I_{SS}}{I_0} \times \frac{\Delta V - I_0 \times R_0}{\Delta V - I_{SS} \times R_{SS}} \tag{4}$$

where $I_0$ and $I_{SS}$ are initial and steady-state current, respectively. $\Delta V$ is the applied polarization voltage. $R_0$ and $R_{SS}$ are the interfacial resistance before and after polarization process, respectively[38,62].

For the galvanostatic intermittent titration technique (GITT) experiments, 0.1 A g⁻¹ current pulses were imposed on the cell for 3600 s, after which the relaxation potentials of the cell were measured for 3600 s when no current was applied. The GITT data have been used to calculate the Li⁺ diffusion coefficients ($D_{Li^+}$) at various voltages using Eq. (5):

$$D_{Li^+} = \frac{4}{\pi\tau} \left(\frac{m_B V_M}{M_B S}\right)^2 \left(\frac{\Delta E_s}{\Delta E_t}\right)^2 \tag{5}$$

in which $m_B$ is the mass of active substance on electrode (g); $M_B$ is molar mass (g mol⁻¹); $V_M$ is molar volume (cm³ mol⁻¹); $S$ is area of electrode plate (cm²); $\tau$ stands for the relaxation time (s), $\Delta E_s$ represents the steady-state potential change via the current pulse, and $\Delta E_t$ is the potential change in current pulse after subtracting the iR drop[63,64].

Si/C@C|PVCM-GPE|Li half battery. The anode electrode was prepared by casting a slurry consisting of active material, Super P, sodium carboxymethylcellulose (CMC), styrene butadiene rubber (SBR) (95.2: 2: 1.4: 1.4, weight ratio) onto copper foil and dried in a vacuum oven at 110 °C overnight. For half-cells with Li metal foil (100 μm) as the counter electrode, the Si/C@C|PVCM-GPE|Li battery that fabricated via in-situ polymerization process, which can be described as follows. Typically, 1 M LiDFOB in VC:EC/DEC (2: 3, v/v) solution with 0.2 wt % AIBN was injected into coin-type battery (CR2032, Guangdong Canrd New Energy Technology Co., Ltd). Nanocellulose (NC) separator was adopted between cathode and anode to prevent the internal short

circuit of the battery before the polymerization of polymer electrolyte. After that, the battery was assembled and transferred to an oven at 60 °C for 24 h and followed by the polymerization at 80 °C for another 2 h to make the complete polymerization of PVCM-GPE. The assembly of all the cells were carried out in a glovebox filled with argon gas ($H_2O < 1$ ppm, $O_2 < 0.1$ ppm).

For reference measurements, batteries with liquid electrodes were also prepared, following the general procedure described above without polymerization procedure, named Si/C@C|LE. The charge/discharge process at 25 °C with a voltage range of 0.01-0.8 V at different current densities was conducted on a NEWARE Battery Test System (CT-4008Tn-5V10mA-164, Shenzhen, China). The GITT measurements were conducted on the NEWARE Battery Test System from 0.01 to 2.0 V.

**Si/C@C-Gr|PVCM-GPE|NMC811 full cell.** As for the pouch full cell, the mixed Si/C@C and commercial graphite (Gr) with designed specific capacity were used as an anode, and the commercial $LiNi_{0.8}Mn_{0.1}Co_{0.1}O_2$ (NMC811) composite electrode was used as a cathode. The cathode electrode was fabricated by mixing NMC811, Super P, and polyvinylidene difluoride (PVDF) (98.2: 1: 0.8, weight ratio) in N-methylpyrrolidone (NMP) into a homogeneous slurry. The slurry was coated onto an aluminum foil and vacuum-dried at 120 °C for 12 h. The capacity ratio of negative to positive electrodes (N/P ratio) in the full cell is about 1.07. The full-cells were assembled in an aluminum-plastic pouch type cell named Si/C@C-Gr|PVCM-GPE|NMC811. Full-cell tests by constant-current constant-voltage charging (CCCV) and constant-current (CC) discharging at different C-rates (1 C = 2.7 A) between 4.2 and 3.0 V. Pre-activation generally involved cycling the cell for 3 cycles at 0.1 C. For the operando XRD analysis, the full cells were assembled by integrating the electrodes in a single layer aluminum pouch cell.

## Numerical simulation modeling

The mathematical principle of the simulation is the finite element analysis method, where a steady state study of the 3D structural model containing parametric scans is analyzed by using the Solid Mechanics Physical Field Interface in the Structural Mechanics module of the COMSOL Multiphysics software. Particular attention should be paid to the fact that the material properties with the lithiation stage. All simulation models were dissected using free tetrahedral meshes to ensure simulation accuracy of the models. The solver is PARDISO[65,66]. For diffusion-induced stress analysis of $Li_xSi$ as a function of Li concentration, Young's modulus from 90 to 40 GPa and Poisson's ratio from 0.28 to 0.24 was adopted. The yield strength of pristine Si is set at 1.75 GPa[67].

The HOMO/LOMO energies were performed using a Materials Studio software package. Equilibrium state structures were fully optimized using the B3LYP method and the 6-311++G (d) basis set[68].

## Data availability

The source data used in this study are available in the Figshare database (https://doi.org/10.6084/m9.figshare.25534249). All other data are available from the corresponding author upon request.

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

## Acknowledgements

The authors appreciate the financial support from the National Natural Science Foundation of China (52173229 and 52373229), the Natural Science Foundation of Shaanxi (2019KJXX-099 and 2023-JC-JQ-15), the Fundamental Research Funds for the Central Universities (3102019JC005) and the Key Research and Development Projects of Shaanxi Province (No. 2019ZDLGY04-05). Furthermore, we would like to thank the Analytical & Testing Center of Northwestern Polytechnical University for providing several testing instruments; First-principles calculations were performed on the High-Performance Computation Center at Northwestern Polytechnical University.

## Author contributions

M.B. conceived the idea, conducted the experiments, and wrote the manuscript. H.W. and A.S. carried out the transition electron microscopy characterization. M.B., X.T., M.Z., Z.W. and Y.M. discussed the results and commented on the manuscript. Y.M. supervised the research.

## Competing interests

The authors declare no competing interests.
