## [Peer Review File · Nature Communications]

REVIEWER COMMENTS

Reviewer #1 (Remarks to the Author):

The development of high-energy-density and long lifespan high Ni cathode/Si-based anode battery is crucial for extending driving range of electric vehicles and alleviating range anxiety among users. Nonetheless, both high Ni cathode and Si-based anodes face severe issues of degradation of interface and disintegration of bulk materials. In this work, Ma and coworkers combined the Si structural engineering and quasi-solid electrolyte strategies to significantly alleviate the cross-talk effect and stress concentration issue of the high-Ni/Si-based battery. The Si/C@C-Gr 550|GPE|NMC full cells demonstrated high energy density, long cycling stability and wide serving temperature range. The following questions should be answered based on major revisions before consideration for publication in prestigious Nature Communication.

1. In introduction, the authors claim that “when using traditional ethylene carbonate electrolyte, the Fermi energy incompatibility at the multi-scale interface will cause the interface impedance to be unregulated”. Similar also descriptions also appear in the authors’ previous published work (Advanced Energy Materials, 2022, 12(32): 2201390). However, no relevant description appears in the references cited in this section. The author is requested to explain this incompatibility in detail. Are there significant incompatibilities at all liquid-solid interfaces, not just the liquid carbonate electrolyte system?

In addition, the article should also add a corresponding discussion on the use of this quasi-solid electrolyte to alleviate the Fermi energy incompatibility at the interface. In fact, the Fermi energy at the solid-solid interface should also be very incompatible.

2. There are too many variables in the simulation part. It is more appropriate to study the stress evolution of the same active material under different electrolytes. In addition, it is inappropriate to use the modulus and Poisson's ratio of pure silicon here. The modulus and Poisson's ratio of the Si/C composite should be used.

3. Similar Si structure engineering has been reported in the authors’ previous work (Nano Energy, 2022, 95: 107026). The highlight of this paper is the proposal of this in-situ quasi-solid electrolyte strategy, as the title expresses. To this end, it is recommended that the authors add electrochemical data of pure silicon as the anode active material to more strongly prove the effectiveness of this electrolyte strategy.

4. The descriptions of current density for full-cell measurement are contradictory in the main text (1 C = 200 mA g⁻¹) and experimental sections (1 C = 100 mA g⁻¹). Also, please give the current density for Si-based anode in half cell.

5. The authors have conducted a lot of work on developing transmission mode operando phase change technology to investigate the structural evolution of electrode materials. It is recommended that the authors give a schematic illustration to visually display the operando XRD measurement of NCM811/Si-based pouch cell during charging/discharging process.

6. Why galvanostatic charge-discharge and GITT tests use different voltage ranges.

Reviewer #2 (Remarks to the Author):

The paper titled "A facile in-situ polymerization strategy for the energy/power-dense Si anode || Ni-rich cathode battery system: cooperative manipulation of stress dissipation and mitigated cross-talk effect" presents a novel approach in battery technology. This research addresses the mechanical instability of Li-Si alloy cathodes and the challenges associated with high-voltage cycling, Li⁺ diffusion kinetics, and leakage currents at elevated temperatures. The paper introduces a gel polymer electrolyte composed of lithium difluoro(oxalate) borate salt and in-situ polymerized poly(vinylene carbonate) matrix (LiDFOB-PVCM). This is a significant innovation in enhancing the mechanical integrity of Si anodes and stabilizing interfacial properties. The research includes extensive operando X-ray diffraction documentation, demonstrating enhanced Li utilization and reduced lattice breathing in layered oxide cathodes. However, before the acceptance of this manuscript, I'd like to see the answers/comments to the questions below:

1. Limited Real-world Application: The study predominantly focuses on laboratory settings and controlled environments. Translating these findings to real-world applications, such as in commercial battery production, may present challenges not addressed in the paper.

2. Scalability and Cost: The materials and processes described, while innovative, may not be easily scalable or cost-effective for mass production. The feasibility of implementing these techniques in large-scale manufacturing is not thoroughly discussed.

3. Long-Term Durability and Performance: While the paper presents promising short-term results, it lacks extensive long-term performance data. This is crucial for assessing the practicality of the technology in commercial batteries that require long lifespans.

4. Environmental and Safety Considerations: The environmental impact and safety aspects of the new materials and processes are not extensively covered. These areas need attention given the increasing emphasis on sustainable and safe battery technology.

5. Comparative Analysis with Existing Technologies: The paper could benefit from a more comprehensive comparison with existing technologies. This would help understand the proposed method's relative advantages and limitations.

Reviewer #3 (Remarks to the Author):

The manuscript reported a gel polymer electrolyte that composed of the lithium difluoro(oxalate) borate salt and the in-situ polymerized poly(vinylene carbonate) matrix (LiDFOB-PVCM), which not only reinforces the mechanical integrity of Si anode (3.5 mA h cm^{-2}), but also chelates with the transitional cations towards the stabilized interfacial property. This is a very interesting and innovative study, the authors provide a logical and systematic presentation from material selection, and synthetic process exploration, to mechanism and application investigations, which is deserving to be shared in the field of high energy density battery area. Overall, this is a well-written and innovative manuscript that is suitable for publication in Nature Communications. Therefore, I highly recommend publishing this paper. Moreover, there are a few minor issues that can still be discussed.

1. Does the 2-amino-2-methyl-1-propanol affect the cycling?
2. How to prove the mechanical sheath effect of pyrolytic layer?
3. Have the authors conducted more investigations on the comparison sample Si/G@C?
4. Please supplement the process and details of the peeling test.
5. What percentage of DEC and EC is in the polymer electrolyte? How DEC and EC residual solvent affect cycling?
6. How crosstalk effects are suppressed by gel electrolytes?
7. Why the Si/C@C mixed with artificial graphite can enhance the compatibility with cathode?

Responses to reviewers' comments

Reviewer #1:

The development of high-energy-density and long lifespan high Ni cathode/Si-based anode battery is crucial for extending driving range of electric vehicles and alleviating range anxiety among users. Nonetheless, both high Ni cathode and Si-based anodes face severe issues of degradation of interface and disintegration of bulk materials. In this work, Ma and coworkers combined the Si structural engineering and quasi-solid electrolyte strategies to significantly alleviate the cross-talk effect and stress concentration issue of the high-Ni/Si-based battery. The Si/C@C-Gr 550|GPE|NMC full cells demonstrated high energy density, long cycling stability and wide serving temperature range. The following questions should be answered based on major revisions before consideration for publication in prestigious Nature Communication.

Response: We greatly appreciate the reviewer's positive evaluation of this manuscript. We have implemented all the changes suggested by the reviewer. The point-by-point responses to the questions are provided below.

1. In introduction, the authors claim that "when using traditional ethylene carbonate electrolyte, the Fermi energy incompatibility at the multi-scale interface will cause the interface impedance to be unregulated". Similar also descriptions also appear in the authors' previous published work (Advanced Energy Materials, 2022, 12(32): 2201390). However, no relevant description appears in the references cited in this section. The author is requested to explain this incompatibility in detail. Are there significant incompatibilities at all liquid-solid interfaces, not just the liquid carbonate electrolyte system? In addition, the article should also add a corresponding discussion on the use of this quasi-solid electrolyte to alleviate the Fermi energy incompatibility at the interface. In fact, the Fermi energy at the solid-solid interface should also be very incompatible.

Response:

Thanks for the reviewer's comments. We really apologize of the unclear description and confusion caused. In retrospect, we should have better clarified the pivotal role of

the quasi-solid electrolyte in regulating the multiscale interfacial stability Admittedly, many studies have indicated the significance of the highest occupied molecular orbital (HOMO) and lowest unoccupied molecular orbital (LUMO) of the liquid electrolyte where either the solvent or the salt oxidation/reduction take place. For example, Goodenough pointed that the voltage window of the electrolyte is the energy gap between its lowest unoccupied and highest occupied molecular orbitals (LUMO and HOMO) of a liquid electrolyte (J. Am. Chem. Soc. 2013, 135, 1167–1176; Energy Environ. Sci. 2014, 7, 14–18.). A fermi level of anode above the electrolyte LUMO reduces the electrolyte; similarly, the fermi level of cathode located below the HOMO oxidizes the electrolyte. As noted in the reference of Adv. Energy Mater. 2022, 12, 32, 2201390.

Fig. (a) Relative energies of the electrolyte window E_g and the electrode electrochemical potentials μ_A and μ_C with no electrode/electrolyte reaction: liquid electrolyte with solid electrodes (J. Am. Chem. Soc. 2013, 135, 1167–1176); (b) Voltage versus capacity of insertion-compound electrodes (Energy Environ. Sci. 2014, 7, 14–18)

The voltage window of commercial carbonate electrolyte is about 1–4.2V. To maximum the energy density of Li-ion batteries, the potential of typical LIB electrode host material exceeds this voltage range. In other words, the electrolyte is thermodynamically unstable upon cycling. The deposition of the reduction product on the anode surface would form the solid electrolyte interphase and insulate the further decomposition of electrolyte. The gap between HOMO/LUMO of the carbonate electrolyte and the fermi level of high-capacity cathode/anode is the issue of the incompatibility we described in the manuscript. This interfacial incompatibility relies

on the different choice of electrolyte and electroactive material. For instance, the ester-based electrolytes demonstrate the better compatibility with the low voltage range of the high-capacity metallic/Si anodes, while the sulfonamide-based electrolyte enables 4.7 V lithium metal battery cycling (Nat. Energy 2021, 6, 495–505). In this sense, the compatibility degree depends on the specific solvent/salt formulations, not only limited to the carbonate electrolyte.

In this study, we proposed the VC additive as the film-forming agent, which decomposes through an electrochemically triggered radical polymerization. The as-formed cross-linked polymeric product effectively improve the robustness of SEI layer (J. Power Sources 2020, 477, 228567). That is why the gel polymer electrolyte could prolong the lifespan of Si/C anode. But there is indeed no evidence to confirm that the quasi-solid electrolyte with a higher LUMO compared with silicon and graphite. We believed the better compatibility is due to the less content of the free solvent molecules after the polymerization in contact with the anode. Consequently, in the revised manuscript, we would delete this sentence in the revised manuscript for the more clear: “the Fermi-energy incompatibility at the multiscale interface would lead to the unregulated interfacial impedance.”

2. There are too many variables in the simulation part. It is more appropriate to study the stress evolution of the same active material under different electrolytes. In addition, it is inappropriate to use the modulus and Poisson’s ratio of pure silicon here. The modulus and Poisson's ratio of the Si/C composite should be used.

Response:

We really appreciate the reviewer for the highlighting this critical issue. Based on the reviewer’s suggestion, we have supplemented models of Si/G@C|GPE and Si/C@C|LE (Fig. S12a) besides Si/G@C|LE and Si/C@C|GPE in Fig. 3. As shown in Fig. S12a, the Si/G@C electrode with GPE (Si/G@C|GPE) exhibited high stress concentrations in the Si particle region during the lithiation process. In sharp contrast, as the Si particles were evenly distributed in the CTP derived pyrolytic carbon, the stress distribution became more homogeneous in the Si/C@C|LE (Fig. S12b). The low stress concentration is attributed to the spatially confined individual Si NPs in the CTP-derived carbon. As

compared in Fig. S12d, the carbon filler increased the interparticle contact area with the progressive stress dissipation, which ensures the mechanical stability of the composite particles during the lithiation. The mechanical stress of Si NPs in the Si/G@C|LE model is estimated ~ 2.1 times larger than in Si/C@C|GPE, and the compressive stress in the carbon layer is ~ 18 times (Fig. S12e). The dissipation of tensile stress ensures the mechanical stability of Si/C@C|GPE upon the lithiation. From the perspective of mechanical constraints, our modeling studies indeed provide insights towards optimization of spatial arrangement for the Si based architectures. With the Si/G@C|LE and Si/C@C|GPE results in Fig. 3, the intrinsic structure of the Si/C particles is more crucial for the anode structural robustness, even though GPE partly mitigate the stress concentration. Based on the suggestion, above discussions have been supplemented into the manuscript.

Fig. S12 Chemomechanical modeling of stress distribution during lithiation. Stress distribution modeling across the anodes for **a** Si/G@C|GPE and **b** Si/C@C|LE at different lithiation states, and stress distribution of the single composite particle at the deep lithiation state. Stress changes of Si NPs and carbon layer of **c** Si/G@C|GPE and **d** Si/C@C|LE at 100%

lithiation state. e Stress comparison of the Si and carbon layer species from two models.

In addition, this work focuses on the mechanical characteristics of silicon and carbon itself, respectively. Silicon expands tremendously during the lithiation process (one order magnitude lower than the amorphous carbon), while the volume expansion of carbon material is relatively almost negligible and the formats of different polymorph and porosity are difficult to be predicted. In this regard, we have to use the Poisson's ratios of only silicon to simplify the rough difference between the mechanical stresses among various models.

3. Similar Si structure engineering has been reported in the authors' previous work (Nano Energy, 2022, 95: 107026). The highlight of this paper is the proposal of this in-situ quasi-solid electrolyte strategy, as the title expresses. To this end, it is recommended that the authors add electrochemical data of pure silicon as the anode active material to more strongly prove the effectiveness of this electrolyte strategy.

Response:

We highly appreciate for reviewer's comments, which inspire us to further polish the manuscript and elaborate on the highlights. As suggested by the reviewer, the following Fig. S15 summarized the electrochemical performance of pure Si electrodes obtained in half-cells as paired with GPE or LE. The cycle behaviors of the Si|GPE and Si|LE anodes were compared at 200 mA g^{-1} with the similar areal capacity loading of $\sim 2 \text{ mg cm}^{-2}$ (Fig. S15a). The first discharge and charge capacities of Si|GPE were documented as $3554.6 \text{ mA h g}^{-1}$ and $3016.6 \text{ mA h g}^{-1}$, rendering a satisfactory initial CE (ICE) of 84.9%. Meanwhile, Si|GPE renders better capacity retention (CR) of 44.1% for 30 cycles and superior average CE value (98.9%) from the 3rd cycle onwards. For comparison, the Si|LE displayed the initial ICE value of 81.6% and 0.7% CR value for 30 cycles. Obviously, the difference in performance is determined by the different characteristics of the electrolyte.

As indicated below, the continuous growth of the solid electrolyte phase (SEI) on the highly active Si NPs is a major source of cation depletion and poor cycle life for the electrode. These deficiencies are exacerbated by a significant volume expansion of

silicon (>300%) during the lithiation process, and the loss of Li^+ inventory due to SEI growth and irreversible trapping of lithium-silicon alloys within the active material (*Science* 2021, 373, 1494–1499). The capacity loss of the battery mainly comes from LAM (loss of active material) and LLI (loss of lithium inventory) (*ACS Appl. Energy Mater.* 2022, 5, 11, 13367–13376; *Phys. Chem. Chem. Phys.* 2021, 23, 8200–8221). In the Si|GPE anode, PVCM-GPE encapsulation acts as a stress buffering layer to alleviate the volume expansion of silicon, which ensures the mechanical stability of the active particles during the lithiation, thus mitigating the active material loss. Furthermore, this structural stability also avoids active Li^+ depletion due to repeated generation of SEI (Fig. S15b). The polymerized VC on the anode with excellent electrolyte percolation capability, is conducive to the electrolyte retention on the anode surface and reduces the negative impact of localized solvent dried-up.

Fig. S15 (a) The cyclability values of the Si|GPE|Li and Si|LE|Li at 200 mA g^{-1} . (b) Schematic illustration of Si|GPE|Li and Si|LE|Li half-cell model.

4. The descriptions of current density for full-cell measurement are contradictory in the main text ($1 \text{ C} = 200 \text{ mA g}^{-1}$) and experimental sections ($1 \text{ C} = 100 \text{ mA g}^{-1}$). Also, please give the current density for Si-based anode in half cell.

Response:

We apologize for the confusion caused. For a full battery with 2.7 A h capacity, 1 C corresponds to a current of 2.7 A . The calculation of C rate is neglected because the current is directly used as an input parameter in the test of software. We have corrected this mistake in the revised manuscript. Regarding the half-cell evaluations, C rate was calculated based on the designed specific capacity of the anode material. For example,

with a silicon-carbon anode average specific capacity of 1320 mA h g^{-1} in Fig. 4a, 1 C was defined as 1320 mA g^{-1} . Based on the suggestion, we've made corrections in the manuscript.

5. The authors have conducted a lot of work on developing transmission mode operando phase change technology to investigate the structural evolution of electrode materials. It is recommended that the authors give a schematic illustration to visually display the operando XRD measurement of NCM811/Si-based pouch cell during charging/discharging process.

Response:

We appreciate for reviewer's comment about this point. As for the referee's concern, we have supplemented a schematic to display the operando XRD measurement in the manuscript Fig. S23. The single layer pouch cell was assembled for in-situ XRD test. The specific process is as follows: firstly, the Si-based anode and NMC811 cathode were cut to $61 * 41 \text{ mm}^2$ and $60 * 40 \text{ mm}^2$ respectively. Secondly, the tabs were welded to the electrodes. The cathode, separator and anode were carefully aligned, and sealed into pouch bag. The fourth side was left unsealed. Then the pouch cell was introduced into glovebox and filed with different electrolyte. Naturally, a battery with a gel electrolyte has to undergo a thermopolymerisation process.

After assembling, the single layer pouch cell was fixed on the sample holder by a clamp. Finally, the pouch cell was charged and discharged while X-ray diffraction spectra were started to be collected. The dynamic phase transition in NMC811/Si-based pouch cell upon the charging/discharging process were identified and documented by a transmission-mode X-ray diffractometer (STADIP STOE) with a position-sensitive detector and Mo $K\alpha_1$ radiation with a wavelength (λ) of 0.70930 \AA , operating at 50 kV and 40 mA.

Fig. S23 Schematic illustration of operando XRD measurement of the Si-based anode||NMC full cell.

6. Why galvanostatic charge-discharge and GITT tests use different voltage ranges.

Response:

We appreciate for reviewer's comment about this point. For the galvanostatic intermittent titration technique (GITT) experiments, 0.1 A g^{-1} current pulses were imposed on the cell for 3600 s, after which the relaxation potentials of the cell were measured for 3600 s when no current was applied. The GITT measurements were conducted from 0.01 to 2.0 V while the galvanostatic tests were conducted between 0.01 to 0.8 V. The higher cut-off voltage would deteriorate the long-term cycling stability due to the deep delithiation stage. In comparison, the wider voltage range utilized in GITT can offer a more comprehensive understanding of the kinetic behavior of lithium ions, surpassing the limitations observed during long-term cycling. Previous works also choose the similar strategy (Sci. Rep. 2020, 10, 14966; Energy Environ. Sci. 2015, 8, 2075–2084).

Reviewer #2:

The paper titled "A facile in-situ polymerization strategy for the energy/power-dense Si anode||Ni-rich cathode battery system: cooperative manipulation of stress dissipation and mitigated cross-talk effect" presents a novel approach in battery technology. This research addresses the mechanical instability of Li-Si alloy cathodes and the challenges associated with high-voltage cycling, Li⁺ diffusion kinetics, and leakage currents at elevated temperatures. The paper introduces a gel polymer electrolyte composed of lithium difluoro(oxalate) borate salt and in-situ polymerized poly(vinylene carbonate) matrix (LiDFOB-PVCM). This is a significant innovation in enhancing the mechanical integrity of Si anodes and stabilizing interfacial properties. The research includes extensive operando X-ray diffraction documentation, demonstrating enhanced Li utilization and reduced lattice breathing in layered oxide cathodes. However, before the acceptance of this manuscript, I'd like to see the answers/comments to the questions below:

Response:

We appreciate for all comments regarding this manuscript. All the suggestions are very valuable, for both polishing this manuscript and guiding the further research. All your questions are answered point by point as below.

1. Limited Real-world Application: The study predominantly focuses on laboratory settings and controlled environments. Translating these findings to real-world applications, such as in commercial battery production, may present challenges not addressed in the paper.

Response:

We really appreciate the reviewer for highlighting this point. Besides the fundamental research progress about synthesizing materials and preparing cells made in laboratory, addressing scientific challenges from a novel perspective is indeed crucial for further large-scale manufacturing:

(i) For the alternative high-capacity anodes, the raw composite material of silicon-based composite mainly consists of silicon and graphite species from the commercial

products. Silicon alloys with lithium at room temperature and has a theoretical specific capacity of up to 4200 mA h g^{-1} , which is more than tenfold of that of the graphite anodes. Besides the high-capacity merits, favorable features also involve the suitable discharge plateau, abundant earth reserves (27.72%) and low price ($\sim \$2,266$ per ton), which collectively render the silicon-carbon anode one of the more promising next-generation anode materials. In this work, the low-priced, highly industrialized pitch material ($\sim \$500$ per ton) was employed. Among various production protocols of silicon-based anode materials, the scalable sand milling approach was selected for its simplicity and cost-effectiveness. The as-obtained Si NPs exhibits small particle size and uniform distribution, which is applicable for the additional powder processing. Additionally, the spray drying and sintering processes, utilized for synthesizing silicon-carbon materials, are well-established industrial techniques.

(ii) As for the cathode materials, high Ni cathode stand out with their high specific capacity, compact density and energy density. The strategic adjustment in the composition of NMC811 cathode material, with increased nickel and reduced cobalt content, not only mitigates the cost challenges associated with cobalt scarcity but also renders a specific capacity exceeding 200 mA h g^{-1} , ensuring its market competitiveness.

(iii) Regarding the electrolyte formulation, the gel electrolyte employed in this work comprises the commercially available lithium salt (LiDFOB) and organic solvents (EC, DEC) and VC additives. Moreover, the whole cell assembly well integrates into the existing cell manufacturing procedures, which only requires the mild-temperature heating step for the in-situ polymerization, with negligible labor or cost input. Based on the reviewer's comment, we have provided this comment of the potential suitability for the commercial battery production in the revised manuscript (material synthesis in experimental section)

2. Scalability and Cost: The materials and processes described, while innovative, may not be easily scalable or cost-effective for mass production. The feasibility of implementing these techniques in large-scale manufacturing is not thoroughly discussed.

Response:

We really appreciate the reviewer for highlighting this crucial point. Silicon exhibits a high earth's crust abundance (27.72%), low price (~\$2,266 per ton), and a theoretical lithium storage capacity of up to 4200 mA h g⁻¹, which is 10 times higher than the capacity of graphite (372 mA h g⁻¹). The voltage plateau of silicon is higher than that of graphite, preventing the metallic deposition and alleviating the short circuit risks. Thus, the safety performance is better than that of graphite anode material. However, the volume expansion of pure silicon material during charging and discharging can reach ~300%, which leads to the severe pulverization of silicon anodes and the repeated regeneration and thickening of SEI film and compromises the initial cycle efficiency and life span of the battery. Therefore, silicon-carbon composites are the ideal route to improve the energy density of batteries. So far, significant industrial efforts have been devoted to the mass production of silicon-based anodes. The mainstream production strategies for silicon NPs includes the mechanical milling (ball milling and sand milling), chemical vapor deposition, as well as the metallic dealloying methods. These techniques are being extensively explored and optimized to enhance the efficiency and scalability of silicon anode production. In this work, the mechanical sand milling equipment capable of handling quantities up to 10 kg raw material was utilized (WSP-10, Longxin Intelligent Equipment Co., Ltd). This approach provides a reliable and consistent Si NPs at a scale suitable for industrial applications. To guarantee superior performance, the stepwise powder processing techniques, including the spray drying and high-temperature pyrolysis were also employed. Actually, all these procedures have progressed to the pilot-line production stage. (high-speed centrifugal spray dryer: LGP-25, Jiangsu Xinma Drying Technology Co., Ltd; rotatable furnace: KY-R, Xianyang Research Design Institute of Ceramics Co., Ltd). In this regard, the consistent and reproducible results could be guaranteed, thereby facilitating the commercialization of this technology.

The increased manufacturing cost, from the liquid batteries to gel electrolytes is relatively low. Notably, the electrolyte perfusion occurs without any alterations to the electrode preparation process, ensuring seamless integration with the commercial cell

production and cost-effectiveness (Nano-Micro Lett. 2024, 16, 2). On the one hand, the lithium salt (LiDFOB) and the organic solvents (EC, DEC, VC) of the gel electrolyte used in this work are all the materials that already available for industrial application (Chem. Mater. 2022, 34, 10, 4587–4601). In this work, the mixed Si/C@C and commercial graphite (Gr) with designed specific capacity were used as anode electrode, and the commercial $\text{LiNi}_{0.8}\text{Mn}_{0.1}\text{Co}_{0.1}\text{O}_2$ (NMC811) composite electrode was used as a cathode for the pouch-format full cell prototyping. The 2.7 A h full-cells were assembled in an aluminum-plastic pouch type cell (Si/C@C-Gr|PVCM-GPE|NMC811). A stable capacity retention of 88.7% was achieved for 2000 cycles (Fig. 6b).

3. Long-Term Durability and Performance: While the paper presents promising short-term results, it lacks extensive long-term performance data. This is crucial for assessing the practicality of the technology in commercial batteries that require long lifespans.

Response:

Thanks for the review's valuable suggestion. For the half-cells, as shown in Fig. 4d, the CR values of Si/C@C-Gr|GPE with predetermined specific capacities of 1000 mA h g^{-1} (denoted as Si/C@C-Gr 1000), 850 mA h g^{-1} (denoted as Si/C@C-Gr 850) and 650 mA h g^{-1} (denoted as Si/C@C-Gr 650) were evaluated as 95.2%, 96.8% and 98.6% for 100 cycles, respectively; the average CEs of all these hybrid anodes were maintained higher than 99.9% after 10 cycles. The capacity retentions of Si/C@C-Gr 1000, Si/C@C-Gr 850, and Si/C@C-Gr 650 cycled up to 200 cycles are illustrated in Fig. a, decreasing to 81.9%, 84.5% and 93.5%, respectively. As the specific capacity increased to 1000 and 1250 mA h g^{-1} , the Si/C@C-Gr 1000|GPE|NMC and Si/C@C-Gr 1250|GPE|NMC pouch-format prototypes can deliver capacity retention of 81.3% and 74.5% after 300 cycles, respectively (Fig. b).

Based on the reviewer's suggestion, the high specific capacity Si/C@C (1250 mA h g^{-1}) was mixed with graphite as an anode material with a specific capacity of 550 mA h g^{-1} , and paired with the NMC cathodes in the commercial full cell configurations, the model of which were thus designated as the Si/C@C-Gr 550|GPE|NMC and Si/G@C-Gr 550|LE|NMC cell models. The Si/C@C-Gr 550|GPE|NMC exhibited an initial charge capacity of 2.69 Ah and discharge capacity of 2.53 Ah, respectively,

corresponding to an ICE of 94.21%. A stable capacity retention of 88.7% was achieved for 2000 cycles (Fig. 6b), in contrast to merely 79.4% capacity retention of the Si/C@C-Gr 550|LE|NMC. Due to time constraints, we could not operate these cells for a longer period of time in our experiments, but a fitting approach was used to answer the reviewer's questions. Based on the results of the cycling tests performed, a capacity decay life models were developed for the Si/C@C-Gr 550|GPE|NMC and Si/C@C-Gr 550|LE|NMC full cells. The cycle life of cells was evaluated by equation $y = a + b \cdot x^c$ (Energy Conversion Congress and Exposition (ECCE), 2015 IEEE, 14–21). The results show that the life of Si/C@C-Gr 550|GPE|NMC is limited to 2875 cycles (80%), and as a comparison, the retention rate of Si/G@C-Gr 550|LE|NMC is reduced to 75.2% (Fig. c). We believe the long cycles is enough to demonstrate the superiority of our proposed in-situ polymerization strategy.

Fig. (a) The cycle performance of the Si/C@C-Gr|GPE anode (blend with graphite) with pre-set nominal capacities of 650, 850, 1000 mAh g⁻¹; (b) Cycling behavior of the Si/C@C-Gr 1000|GPE|NMC and Si/G@C-Gr 1250|GPE|NMC models. (c) Cycling behavior and curves fitting of the Si/C@C-Gr 550|GPE|NMC and Si/C@C-Gr 550|LE|NMC models.

4. Environmental and Safety Considerations: The environmental impact and safety aspects of the new materials and processes are not extensively covered. These areas need attention given the increasing emphasis on sustainable and safe battery technology.

Response:

We appreciate for reviewer's comment about the concerns of "Environmental and Safety Considerations". Regarding to the raw materials, both of the silicon and graphite are abundant in the nature. The extraction processes for silicon and graphite have matured within the industry, which has led to a relatively limited environmental impact. The Si NPs production may consume significant energy. The source of energy (renewable vs. non-renewable) decides the overall environmental footprint. Disposal of silicon/graphite anode materials at the end of their lifecycle is also a concern. While the materials themselves are non-toxic, thus their disposal does not pose any direct risks. But lithium-ion batteries contain other components such as cathode and electrolyte, which should be handled carefully.

Regarding the synthesis process, the solvent medium used in the ball milling process is alcohol, which is recyclable and eco-friendly. The waste gas generated during the calcination process is H₂O, CO, CO₂, H₂, and CH₄ derived from coal tar pitch, which is safe to the environment.

Our anode material and electrolyte design own higher safety features compared to traditional Li-ion battery configuration. Firstly, the discharge plateau of silicon/graphite is higher than pure graphite anode, preventing from lithium dendrite formation. Secondly, The PVCM-GPE demonstrated fire-retardant property as shown in Fig. S11, due to the formation of dense carbon layer on the surface. The usage of gel electrolyte essentially improved the thermal stability of the battery. Thirdly, gel electrolyte can tolerate the mechanical abuse, such as puncture or physical damage, minimizing the risk of short circuits and thermal hazards.

Fig. S11 PVCM-GPE was experienced combustion experiment for three times.

5. Comparative Analysis with Existing Technologies: The paper could benefit from a more comprehensive comparison with existing technologies. This would help understand the proposed method's relative advantages and limitations.

Response:

We highly appreciate for reviewer's comments to help polish the manuscript and elaborate on the highlights. As suggested by the reviewer, we have compared our method with previous research in Fig. 6d and supplemented a compared result as below Table. It can be observed that our work displays the highest energy density and extraordinary long-term stability. The impressive performance was clearly benefitted from the collective effects of the anode structural design as well as the gel electrolyte formulation. Notably, all the performance results were obtained from a 2.7 Ah full cell model of Si/C@C-Gr|PVCM-GPE|NMC811, which further demonstrates the potential practicability of this strategy.

Table. The comparison of the cell behaviors reported in previous studies.

Anode	electrolyte	Energy density of full cell	Cell format	Cycle stability	citation
graphite	poly(vinylene carbonate)		CR-2016 coin cell		1
SiO	elastic gel polymer electrolyte	3.0 mAh cm ⁻²	CR-2032 coin cell	70.0% @350	2
100 nm Si	cyanoresin organogel		CR-2016 coin cell	75% @150	3
hybrid hard carbon-silicon/carbon	1 M LiPF ₆ with 1.0 wt.% VC and 1.0 wt.% LiPO ₂ F ₂ in EC:FEC:EMC = 1:1:6 (m/m/m)	283.9 Wh kg ⁻¹	CR-2025 coin cell	78% @200	4
Micro-sized silicon	1 M LiPF ₆ in FEC/tetramethylene sulfone/1, 1, 2, 2-tetrafluoroethyl 2, 2, 3, 3-tetrafluoropropyl ether = 20:60:20 (v/v/v)	4.1 mAh cm ⁻²	100 mAh pouch cell	89% @120	5
amorphous Si nanodots in carbon nanospheres	1 M LiPF ₆ with 5 wt% FEC in EC/DEC/DMC =1:1:1 (v/v/v)	322.2 Wh kg ⁻¹	Pouch cell	78.3% @1000	6
Si with graphene oxide sheets	1 M LiPF ₆ in EC/DEC = 1:1 (v/v), with 10% FEC	2.14 mAh cm ⁻²	CR-2032 coin cell	70% @100	7
Micro silicon	sulfide solid electrolyte	2 mAh cm ⁻²	Module for solid state electrolyte	80% @500	8
Micron silicon	1 M LiPF ₆ in γ -Butyrolactone/D EC/FEC = 45:45:10 (v/v/v)	5.0 mAh cm ⁻²	400 mAh pouch cell	83.7% @150	9
Si/C@C-Gr	LiDFOB-PVCM	325.9 Wh kg ⁻¹	2.7 Ah Pouch cell	88.7% @ 2000	Our work

Reference

-
1. Chai, J. *et al.* A Superior Polymer Electrolyte with Rigid Cyclic Carbonate Backbone for Rechargeable Lithium Ion Batteries. *ACS Appl. Mater. Interfaces* **9**, 17897–17905 (2017).
 2. Cho, Y.-G. *et al.* Organogel electrolyte for high-loading silicon batteries. *J. Mater. Chem. A* **4**, 8005–8009 (2016).
 3. Huang, Q. *et al.* Supremely elastic gel polymer electrolyte enables a reliable electrode structure for silicon-based anodes. *Nat. Commun.* **10**, 5586 (2019).
 4. Cheng, H. *et al.* High Voltage Electrolyte Design Mediated by Advanced Solvation Chemistry Toward High Energy Density and Fast Charging Lithium-Ion Batteries. *Adv. Energy Mater.*, 2304321 (2024).
 5. Li, A.-M. *et al.* High voltage electrolytes for lithium-ion batteries with micro-sized silicon anodes. *Nat. Commun.* **15**, 1206 (2024).
 6. Li, Z., Han, M., Yu, P., Lin, J. & Yu, J. Macroporous Directed and Interconnected Carbon Architectures Endow Amorphous Silicon Nanodots as Low-Strain and Fast-Charging Anode for Lithium-Ion Batteries. *Nano-Micro Lett.* **16**, 98 (2024).
 7. Sun, B. *et al.* Biomimetics-Inspired Architecture Enables the Strength–Toughness of Ultrahigh-Loading Silicon Electrode. *Adv. Funct. Mater.*, 2314058 (2024).
 8. Tan, D. H. S. *et al.* Carbon-free high-loading silicon anodes enabled by sulfide solid electrolytes. *Science* **373**, 1494–1499 (2021).
 9. Tian, Y.-F. *et al.* Tailoring chemical composition of solid electrolyte interphase by selective dissolution for long-life micron-sized silicon anode. *Nat. Commun.* **14**, 7247 (2023).

Reviewer #3:

The manuscript reported a gel polymer electrolyte that composed of the lithium difluoro(oxalate) borate salt and the in-situ polymerized poly(vinylene carbonate) matrix (LiDFOB-PVCM), which not only reinforces the mechanical integrity of Si anode (3.5 mA h cm^{-2}), but also chelates with the transitional cations towards the stabilized interfacial property. This is a very interesting and innovative study, the authors provide a logical and systematic presentation from material selection, and synthetic process exploration, to mechanism and application investigations, which is deserving to be shared in the field of high energy density battery area. Overall, this is a well-written and innovative manuscript that is suitable for publication in Nature Communications. Therefore, I highly recommend publishing this paper. Moreover, there are a few minor issues that can still be discussed.

Response:

We profoundly appreciate the reviewer's suggestions. We have diligently reviewed all the comments and advice. A point-by-point response has been provided as below.

1. Does the 2-amino-2-methyl-1-propanol affect the cycling?

Response:

We really appreciate the reviewer for highlighting this point. The addition of 2-amino-2-methyl-1-propanol (AMP) is necessary to avoid the nano Si particle agglomeration. The weight ration of AMP to Si is only 0.5% (*Fluid Phase Equilibria* 2002, 198, 239–250). Featured with the positively charged ammonium cation, as a result, the zeta potential of the Si NPs (1M was evaluated as -5.7 mV) was altered to $+18.6 \text{ mV}$ (1M), enabling the monodispersity of Si NPs in the solution due to electrostatic repulsion. In the subsequent spray drying process (process III), the Si NPs were homogeneously dispersed within the pyrolytic CTP at the completion (Supplementary Fig. 2). Considering that the obtained Si/C composite would be heat treated at $920 \text{ }^\circ\text{C}$ after spraying drying process, the 2-amino-2-methyl-1-propanol ($\text{C}_4\text{H}_{11}\text{NO}$) would decompose to C_3H_6 , NH_3 and H_2O over $300 \text{ }^\circ\text{C}$. We believe the AMP does not exist in the final product, therefore, there is no effect on the electrochemical performance of the

battery.

Fig. 1 (a) The sand-milling process of the micro size Si to the Si NPs with the aid of the AMP-95.

2. How to prove the mechanical sheath effect of pyrolytic layer?

Response:

We really appreciate the reviewer for highlighting this point. Regarding the mechanical property of pyrolytic carbon layers, we think this effect can be demonstrated by the following points: (1) Mechanistic aspects: Pitch has a high carbon content and coking residual carbon value. During the calcination process, the pitch pyrolyzes and condenses to form cohesive coke with a high residual carbon content. This cohesive coke has a high binding force and is capable of firmly bonding the carbonaceous aggregate into a compact whole. The physicochemical properties of pitch after carbonization are similar to bones and have high mechanical strength (*Soft Matter*, 2021, 17, 8925-8936; *Diamond and Related Materials*, 2019, 95, 83-90; *Journal of Analytical and Applied Pyrolysis*, 2018, 134, 293-300).

(2) Experimental performance: As shown in Supplementary Fig. 21, the thickness of the Si/C@C-Gr 550|GPE electrode increased from 110 μm (double-side coated, comprising 12 μm Cu foil thickness) to 126 μm at 100% state of charge (SOC) after 100 cycles, corresponding only $\sim 17\%$ volume expansion. This structural robustness reaffirmed the mechanical protection of the CTP pyrolytic carbon and elastic GPE to bear with the deep lithiation. To quantitatively assess the combined effects of GPE and Si/C@C composite design on mitigating volume expansion of the electrode, cross-sectional FESEM images of Si/C@C-Gr 550|GPE, Si/C@C-Gr 550|LE, and Si/G@C-Gr 550|LE electrodes were compared before and after cycling. From the optical photograph in Supplementary Fig. 21, the Si/C@C-Gr 550|GPE electrode was flat, without any notable fractures on the surface. On the contrary, the mechanical strain of

anode composite induces microcracks in Si/C@C-Gr 550|LE electrode, which also demonstrated 19% increase in the thickness at the lithiated state over 100 cycles. Worst of all, the Si/G@C electrode exhibits a 38% increase in thickness owing to the volume expansion of the aggregated Si without adequate reversed compression, leading to obvious electrode delamination upon cycling.

Fig. S21 Cross-sectional and top views FESEM images of the Si/C@C-Gr 550 electrode with GPE and LE before and after 100 cycles. Inset images are the optical photographs of the post-mortem Si/C@C-Gr 550|GPE and Si/C@C-Gr 550|LE electrodes, and summary of electrodes thickness variations.

(3) Simulation results: Based on material design strategies and the mechanical properties afforded by GPE, chemomechanical models were constructed via COMSOL to estimate the mechanical stress distribution upon lithiation. As shown in Fig. 3a, the Si/G@C electrode with LE (Si/G@C|LE) exhibited high stress concentrations in the Si particle region during the lithiation process. The interaction of the accumulated Si NPs squeezed each other upon the deeper lithiation process, meanwhile the edges of Si particles were severely pulverized, eventually leading to the collapse of the electrode structure. The tensile stress and reverse compression at the point–point contact of the aggregated Si NPs suggest that insufficient In sharp contrast, as the Si particles were evenly distributed in the CTP derived pyrolytic carbon, the stress distribution became more homogeneous in the Si/C@C|GPE (Fig. 3b). The low stress concentration is attributed to the spatially confined individual Si NPs in the CTP-derived carbon. As compared in Fig. 3d, the carbon filler increased the interparticle contact area with the progressive stress dissipation. Compared to the Si/G@C|LE structure, the Si/C@C|GPE model is estimated to reduce ~ 47% mechanical stress, while the compressive stress of the carbon layer is reduced by ~ 11% (Fig. 3e). The dissipation of tensile stress ensures the mechanical stability of Si/C@C|GPE upon lithiation. From the perspective of

mechanical constraints, our modeling studies provide insights towards optimization of spatial arrangement for the Si based architectures.

Fig. 3 Stress distribution modeling across the anodes for **a** Si/G@C|LE and **b** Si/C@C|GPE at $\text{Li}_{22}\text{Si}_5$ state. **c** Stress comparison of the Si and carbon layer species from two models.

3. Have the authors conducted more investigations on the comparison sample Si/G@C?

Response:

Thanks for the review's valuable suggestion. For the conventional silicon-graphite hybrid structure of composite anode materials (Si/G@C), we also supplemented several experiments. As shown in Fig. a, the SSA (based on the Brunauer-Emmett-Teller (BET) multilayer adsorption model) of Si NPs with D50 of 127 nm and 137 nm exceeds $50.2 \text{ m}^2 \text{ g}^{-1}$ and $1.8 \text{ m}^2 \text{ g}^{-1}$ for the corresponding Si/G@C composite. More electroactive surface might suffer larger irreversible losses and, hence, lower Coulombic efficiency (CE). It should be noted that although the initial CE (ICE) and average CE of Si/G@C decrease with increasing Si NPs size, this trend does not extend to cycle life. As a result, Si/G@C composite with 165 nm Si NPs achieved optimum CE of 99.57% and cycle life of 114 cycles (cycles number with capacity retention of 80%) at capacity of 705 mA h g^{-1} (Fig. b). This is related to the inability of the structure of this composite anode to perfectly encapsulate the Si particles, resulting in a significant exposure of the Si particles to the electrolyte leading to electrolyte depletion and deactivation of the active substances.

Fig. (a) The cyclability and capacity retention (CR) values of the Si/G@C anodes with different size (127 nm ~ 175 nm) of Si at 0.5 C. **(b)** SSA of Si and Si/G@C, electrochemical performance including cycle life, ICE and average CE of Si/G@C with different size of Si NPs.

3. Please supplement the process and details of the peeling test.

Response:

We highly appreciate for the reviewer for this comment. Peeling tests and tensile experiments were carried out on CMT4204 electronic universal material testing machine. Specifically, the anode slurry was coated on the roughness side of the copper foil (Guangdong Canrd New Energy Technology Co., Ltd). The anodes were cut into 1 × 3 cm² specimens and fixed on the stage of machine with clamp. The adhesive tape (7413D, 3 M) was attached on the side of the electrode with active material. The peeling speed was 100 μm s⁻¹, and the peeling strengths during the detachment were recorded.

Fig. The inset image in Fig. 4h: digital photographs of electrodes after the peeling test.

5. What percentage of DEC and EC is in the polymer electrolyte? How DEC and EC residual solvent affect cycling?

Response:

As shown in Fig. S9b, the two small exothermic peaks centered at 120 °C and 148 °C, correspond to the evaporation of residue DEC and EC, respectively. And the total amount of residual solvent is ~24 wt%.

The presence of residual carbonate solvent promotes the binding of lithium ions

(Li⁺) and carbonyls, which further can enhance mobility of lithium ions within the electrolyte matrix. Residual solvents can wet the interface between the electrode and the electrolyte, increasing lithium transfer rate. Besides, the oxidation potential of EC is typically in the range of 4.0 to 4.2 V vs. Li/Li⁺, lower than poly(VC). Overall, the existence of residual carbonate solvent improved the rate capability of battery, but it may compromise cycle stability.

Fig. S9 (b) TG and DSC of PVCM-GPE with 40% VC.

6. How crosstalk effects are suppressed by gel electrolytes?

Response:

We highly appreciate for the reviewer for this comment. The poly VC contains functional carbonyl group, which consists of a carbon atom double-bonded to an oxygen atom (-C=O). The oxygen atom acts as a donor atom to the metal ion. The electron pair on the oxygen atom in the carbonyl group can form a coordinate covalent bond with transition metal ion. This bond formation involves the lone pair of electrons on the oxygen atom of the carbonyl group being donated to the metal ion, forming a new bond. Besides, ODFB anion captured the dissolution metal ions and forms a stable surface film to prevent the continuing dissolved (*Electrochem. Energ. Rev.* 2019, 2, 128–148; *Adv. Energy Mater.* 2018, 8, 1702561).

We have proved the gel electrolyte can suppress the transitional metal dissolution through XPS, ICP and TOF-SIMS. For the fully delithiated anode as paired with the GPE, the SEI of the anode exhibits much weaker Ni signals than the anode paired with pristine LE (Fig. 5e). Accordingly, the inductively coupled plasma mass spectrometry (ICP-MS) analysis showed that 226 ppm of Ni ion and 118 ppm of Mn ion were

observed at the Si/C@C anode interface in contact with the LE; while only 43 ppm of Ni ion and 27 ppm of Mn ion were observed at anode/GPE electrode, which illuminated that the largely suppressed cross-over effect of TM dissolution from the NMC cathode (Fig. 5f). Meanwhile, the cross-over of the dissolved TM^+ on the surface of the Si/C@C anode electrode is effectively reduced when pairing with the GPE, as evidenced by much weaker $^{58}\text{NiF}_3^-/\text{MnF}_3^-/\text{CoF}_3^-$ signals of the SEI layer. Consistent with the ICP-MS results, this phenomenon further validated the protective role of LiDFOB-PVCM-derived CEI for the cathode (Fig. 5g).

Fig 5. Characterization of CEI and SEI in the Si/C@C|GPE|NMC811 full cell. **c** Proposed model of the SEI and CEI structure in the GPE on electrodes surfaces. **d** Chelation mechanism of TM cations with the LiDFOB salt involved in CEI layer. **e** Ni 2p XPS spectra, **f** ICP-MS analysis of the Si/C@C anode obtained from the Si/C@C-Gr|GPE|NMC811 and Si/C@C-Gr|LE|NMC811 models after 500 cycles.

7. Why the Si/C@C mixed with artificial graphite can enhance the compatibility with cathode?

Response:

We really appreciate the reviewer for highlighting this point. On one hand, as shown in Fig. 4a, although Si/C@C|GPE renders the 95.5% capacity retention and 99.91% average Coulombic efficiency, the initial Coulombic efficiency of Si/C@C paired with GPE is only 83.94%. That means Si/C@C would consume too much Li^+ source from the cathode and lean electrolyte, compromising the practical energy density and cycle life of battery. Artificial graphite anodes typically have an initial Coulombic efficiency in excess of 90%, the additional graphite could improve the initial Coulombic efficiency

and capacity retention. On the other hand, according to Table S2, replacing Si/C@C with Si/C@C-Gr-550 only lose ~10% energy density based on the total mass of pouch cell while the cycling life has been greatly prolonged. Therefore, it is necessary to mix Si/C@C with artificial graphite for constructing full cell.

Fig. 4 Electrochemical performance of the Si-based composite anodes. (a) The cyclability and capacity retention (CR) values of the Si/C@C|GPE, Si/C@C|LE, Si/G@C|LE anodes at 0.5 C. (b) The first charge/discharge, ICE, and average CE of the Si/C@C|GPE, Si/C@C|LE, Si/G@C|LE anodes at 0.5 C.

REVIEWERS' COMMENTS

Reviewer #1 (Remarks to the Author):

The authors have carefully addressed all the previous concerns. Now the manuscript can be accepted for publication.

Reviewer #2 (Remarks to the Author):

The authors have satisfactorily answered all the questions and it is now ready for publication.

Reviewer #3 (Remarks to the Author):

The authors have carefully revised the manuscript and addressed my concerns, I think the manuscript is ready for NC.